# Bayesian Transferability Assessment for Spiking Neural Networks

**Haiqing Hao**                                                    *hhq23@mails.tsinghua.edu.cn*
*Department of Precision Instrument*
*Tsinghua University*

**Wenhui Wang**                                                    *wwh@tsinghua.edu.cn*
*Department of Precision Instrument*
*Tsinghua University*

**Reviewed on OpenReview:** *https://openreview.net/forum?id=GaUtrgXMHe*

## Abstract

Brain-inspired spiking neural networks (SNNs) attract broad interest in neuromorphic computing but suffer the problem of being difficult to optimize. Concurrently, pre-trained models (PTMs) have become a foundation for developing and applying artificial intelligence. Therefore, it is expected that pre-trained SNNs can alleviate the optimization difficulty of training from scratch. However, with a lot of PTMs available in the model hubs, effectively selecting the most appropriate PTM for a given task remains a significant challenge, often necessitating exhaustive fine-tuning and grid-searching. While several solutions to this challenge have been proposed for the mainstream artificial neural network (ANNs), aimed at developing efficient methods to assess the transferability of PTMs on target tasks, the realm of SNNs remains unexplored. The currently most used transferability assessment method for ANNs predicts transferability in a Bayesian perspective. Feature maps extracted by the PTM backbone on the target task are used to calculate the maximum model evidence as the indicator of transferability. However, ANNs and SNNs differ in architecture, rendering the existing Bayesian method incompatible with SNNs. To solve this problem, this paper introduces a novel approach to using the feature maps averaged over the time domain to calculate maximum evidence. Our proposed **M**aximum **E**vidence method with **A**veraged **F**eatures (MEAF) demonstrates effectiveness for SNNs. Additionally, the current algorithm calculates maximum evidence in an iterative way. To accelerate the selection of PTMs, an approximation method is proposed to avoid iteration in the calculation of maximum evidence, significantly reducing time consumption. It is shown through experiment that the proposed MEAF method is effective for the transferability assessment of SNNs. MEAF outperforms information theory-based assessment methods such as LEEP and NCE, which can directly adapt to SNNs on neuromorphic datasets, underscoring its potential to streamline PTM selection and application in the realm of SNNs.

## 1 Introduction

Spiking neural networks (SNNs), known as the third generation of neural networks (Maass, 1997), have obtained broad attention in the field of neuromorphic computing (Roy et al., 2019). Inspired by the way biological neurons communicate with spikes in the brain, neurons in SNNs encode information into sequential binary spikes, offering a potential way towards energy-efficient machine intelligence (Dampfhoffer et al., 2022). Based on its distinctive mechanism, SNNs have achieved outstanding performance in robotics (Bing et al., 2019; Debat et al., 2021), event-based vision (Zhu et al., 2022; Hagenaars et al., 2021), and other fields (Pei et al., 2019; He et al., 2024). However, SNN models are much harder to optimize compared to ANNs (Wu et al., 2019; Wang et al., 2020) because of the non-differentiable nature of spiking signals.

Pre-trained models (PTMs) have fueled the development and applications of artificial intelligence (Jiang et al., 2022). In computer vision, neural networks pre-trained on ImageNet (Deng et al., 2009) exhibit improved performance or convergence efficiency (Yosinski et al., 2014; He et al., 2019; Kornblith et al., 2019), and recent advances in vision-language pre-training provide large-scale visual encoders capable of even zero-shot generalization on downstream tasks (Radford et al., 2021; Jia et al., 2021). Meanwhile, language models self-supervised pre-trained on a large-scale unlabeled corpus demonstrate in-context learning capabilities (Devlin et al., 2018; Brown et al., 2020; Radford et al., 2019; Du et al., 2022). The pre-training and adaptation paradigm has become a foundational methodology for the implementation of large neural network models. With lots of PTMs available, model hubs such as TorchVision (maintainers & contributors, 2016) and Hugging Face Transformers (Wolf et al., 2020) have attracted widespread interest. Meanwhile, pre-trained SNNs have been shown to exhibit higher training efficiency and performance on downstream tasks, significantly reducing the time required to optimize the SNN and improving generalization(Lee et al., 2018; Lin et al., 2022). It is expected that pre-trained SNNs can alleviate the difficulty of optimization when training SNNs from scratch.

However, when using PTMs in a model hub, selecting the most suitable PTM for a given task without exhaustive fine-tuning and grid-searching (You et al., 2021) remains a challenge. Several model hub scheduling methods have been proposed to address this challenge for the mainstream artificial neural network (ANN) (Tran et al., 2019; Nguyen et al., 2020; You et al., 2021). They focus on designing efficient methods to assess the transferability of PTMs on a target dataset and then select the highest-ranked candidate. To the best of our knowledge, however, there is no research focusing on the scheduling of SNN PTMs. Therefore, to bridge the gap, it is meaningful to explore model hub scheduling methods for SNN PTMs.

One problem is that not all existing transferability assessment methods for ANNs can be applied to SNNs directly. Current methods use the pseudo-output (Tran et al., 2019; Nguyen et al., 2020) or the feature map (You et al., 2021; 2022) generated by the PTMs to assess the transferability on the target task. The pseudo-output approaches, which are information theory-based, including LEEP (Nguyen et al., 2020) and NCE (Tran et al., 2019), can be applied directly to SNN. However, the feature-based approach logME (You et al., 2021), which is more effective on ANNs, cannot be adopted to SNN directly. In this logME approach, feature maps extracted by the PTM backbone are used to calculate the maximum model evidence as an indicator of transferability from a Bayesian perspective. Differences between ANNs and SNNs in architecture render the feature-based methods incompatible with SNNs from two aspects. i) The feature map extracted by the SNN backbone has an additional temporal dimension, which cannot adapt to the current model evidence calculation framework. ii) SNNs generate predictions based on the feature with a nonlinear transform, while the maximum model evidence method for ANNs supposes a linear hypothesis.

Taking these two factors into consideration, this paper introduces a Bayesian method of PTM transferability for SNNs inspired by the counterpart for ANNs. To match the feature map of SNN with the current calculation framework of maximum model evidence, we propose to use feature maps averaged through the time domain for the assessment of transferability. Theoretical analysis shows that the nonlinear transform of the SNN layer can be estimated by a linear transform with the proposed averaged features, thus satisfying the linear assumption of the maximum model evidence method. Moreover, the calculation of maximum model evidence requires MacKay's algorithm (MacKay, 1999) to maximize the model evidence iteratively. To further accelerate the selection of PTMs, we propose an approximated maximum model evidence method to avoid iteration while giving similar results as logME. Experiment results show that the proposed **M**aximum **E**vidence method with **A**veraged **F**eatures (MEAF) is effective for SNNs and outperforms information theory-based NCE and LEEP on neuromorphic datasets, and achieves comparable results on static datasets. The effectiveness of the approximated maximum model evidence method is also validated. Our contributions are as follows:

1. We identify the gaps of current Bayesian transferability assessment methods between ANNs and SNNs, and propose the **M**aximum **E**vidence method with **A**veraged **F**eatures (MEAF) for the assessment of transferability of pre-trained SNNs.

2. To further accelerate the selection of PTMs, we provide an approximated maximum model evidence method, which avoids iteration while giving effective results.

3. We validate our proposed method through experiments, and confirm that our Bayesian-based method outperforms information theory-based methods like NCE and LEEP for SNNs on neuromorphic datasets, and achieves comparable results on static datasets.

## 2 Related works

### 2.1 Bayesian model selection

Bayesian model selection methods use model evidence (or marginal likelihood) among different hypotheses as the criterion for selection (Wasserman, 2000). It is believed that model evidence implicitly incorporates the Occam Razor Principle, which balances the complexity and the representation capability of a given model (Knuth et al., 2015). Gull uses model evidence in image reconstruction to determine a critical prior hyperparameter (Gull, 1989). Inspired by Gull's work, model evidence is used in the field of machine learning by MacKay to determine the degree of polynomials in interpolation problems (MacKay, 1992a). It is also used to get the optimal regularization coefficient in the training of regression (MacKay, 1992c) and classification (MacKay, 1992b) problems, even in multi-layer backpropagation neural network models, for better generalization performance (Wolpert, 1992). An algorithm to automatically determine the regularization coefficient in linear regression problems is proposed by MacKay by maximizing the model evidence iteratively (MacKay, 1999).

Evidence-based Bayesian method provides a practical framework to automatically tune the hyperparameters of deep neural networks without the use of validation data. However, a main obstacle lies in the difficulty of estimating the model evidence when the model is deep, as conventional methods proposed by MacKay are only suitable for basic linear regression problems. The Hessian matrix of loss with respect to the trainable parameters is required to calculate the model evidence, which is impossible to efficiently calculate for large-scale neural networks. To solve this problem, scalable Laplace approximation is used to calculate model evidence for deep neural networks (Daxberger et al., 2021; Immer et al., 2021). However, this process is still time-consuming and suffers the problem of inaccurate estimation (Kunstner et al., 2019).

### 2.2 Transferability assessment

Assessment of the transferability of PTMs starts from measuring the similarity between the source task where the models are pre-trained and the target task to transfer. Several distance-based metrics are proposed to measure the difference between source and target tasks, which gives error bounds of the transferred model in theory (Ben-David & Schuller, 2003; Ben-David et al., 2006; Mansour et al., 2009). While it is obvious that models trained on more challenging datasets have better performance when transferred to simple tasks, the opposite is not true. Therefore, distance, which is a symmetry metric, is not an ideal measure of transferability (Nguyen et al., 2020).

As an improvement, some new metrics other than distance have been proposed. Tran et al. (2019) uses conditional entropy (NCE) to estimate the test set accuracy after transfer by calculating the conditional entropy of the target task relative to the source task. Conditional entropy is calculated based on the probability distributions to generate source and target tasks. It is asymmetrical and can reflect the difficult relationship between tasks, achieving good results in experiments.

Contrary to the above methods, Nguyen et al. (2020) no longer uses the relation between tasks as the basis for evaluation, but instead directly inputs the target tasks into the pre-trained model, and quantitatively evaluates the transferability through the output of the pre-trained model. They calculate the expected probability of the real result obtained from the pre-trained model based on the joint distribution between the results obtained from the pre-trained model and the real result of the target task, as a basis for evaluation.

Model evidence-based Bayesian methods are also used in transferability assessment. Kim et al. (2016) uses model evidence to select pre-trained convolutional neural network (CNN) models for best feature extraction. You et al. (2021; 2022) use model evidence to predict the transferability of different PTMs on a given target dataset. In this approach, features extracted by the PTMs instead of predicted results are used as the basis for transferability evaluation. Compared to prediction results, the features extracted from the middle

layer are more universal, so this method can be applied not only to classification problems but also to other problems such as regression.

## 3 Methodology

### 3.1 Background

We consider a supervised learning task on observed data $\mathcal{D} = \{(\boldsymbol{x}_i, t_i)\}_{i=1}^N$ with inputs $\boldsymbol{x}_i$ and targets $t_i \in \mathbb{R}$. SNN model $\mathcal{M}_{\boldsymbol{\theta}}$ predicts $\hat{t}_i$ for each input $\boldsymbol{x}_i$, where $\boldsymbol{\theta} \in \mathbb{R}^D$ represents the trainable weights of the model. The model evidence of $\mathcal{M}$ on task $\mathcal{D}$ is defined as the probability the model generates the observed data, which is calculated as

$$p(\mathcal{D}|\mathcal{M}) = \int_{\mathbb{R}^D} p(\mathcal{D}, \boldsymbol{\theta}|\mathcal{M})\mathrm{d}\boldsymbol{\theta}. \tag{1}$$

For different candidate models, the best one can be chosen by maximizing the model evidence in the Bayesian perspective.

The model evidence can be used to automatically determine the regularization coefficients in ridge regression tasks. Given feature vectors $\boldsymbol{f}_i, i = 1, \cdots, N$, the model predicts $\hat{t}_i, i = 1, \cdots, N$ using a linear transformation. Ridge regression assumes that the parameters follow a Gaussian prior distribution $\boldsymbol{\theta} \sim \mathcal{N}(0, \alpha^{-1}\mathbf{I})$, where $\alpha$ controls the strength of regularization. Furthermore, it assumes that the targets $t_i$ are conditionally distributed according to a Gaussian likelihood $t_i \sim \mathcal{N}(\boldsymbol{\theta}^{\mathrm{T}}\boldsymbol{f}_i, \beta^{-1})$, where $\beta^{-1}$ represents the noise variance. For determination of regularization coefficients, linear regression models with different regularization coefficients $\alpha, \beta$ are viewed as different candidate models, and the evidence is calculated as a function of $\alpha, \beta$ like

$$\begin{aligned} Evidence(\alpha, \beta) &= p(\mathcal{D}|\alpha, \beta) \\ &= \int_{\mathbb{R}^D} p(\mathcal{D}|\boldsymbol{\theta}, \beta)p(\boldsymbol{\theta}|\alpha)\mathrm{d}\boldsymbol{\theta} \\ &= \left(\frac{\beta}{2\pi}\right)^{\frac{N}{2}} \alpha^{\frac{D}{2}} \exp\left(-\frac{\beta}{2}\|\mathbf{F}\boldsymbol{m} - \boldsymbol{t}\|^2 - \frac{\alpha}{2}\boldsymbol{m}^{\mathrm{T}}\boldsymbol{m}\right) (\det \mathbf{A})^{-\frac{1}{2}}, \end{aligned} \tag{2}$$

where $\mathbf{F} \in \mathbb{R}^{N \times D}$ denotes inputs and $\boldsymbol{t} \in \mathbb{R}^N$ denotes targets. $\mathbf{A} = \alpha\mathbf{I} + \beta\mathbf{F}^{\mathrm{T}}\mathbf{F}$, $\boldsymbol{m} = \beta\mathbf{A}^{-1}\mathbf{F}^{\mathrm{T}}\boldsymbol{t}$.

For convenience of calculation, the logarithm of equation 2 is more often used as

$$\mathcal{L}(\alpha, \beta) = \frac{N}{2}\log\beta - \frac{N}{2}\log 2\pi + \frac{D}{2}\log\alpha - \frac{\beta}{2}\|\mathbf{F}\boldsymbol{m} - \boldsymbol{t}\|^2 - \frac{\alpha}{2}\boldsymbol{m}^{\mathrm{T}}\boldsymbol{m} - \frac{1}{2}\log\det\mathbf{A}. \tag{3}$$

We denote $\gamma = \sum_{i=1}^D \frac{\beta\sigma_i}{\alpha+\beta\sigma_i}$, where $\sigma_i$ is the $i_{th}$ singular value of $\mathbf{F}^{\mathrm{T}}\mathbf{F}$.

By iteratively updating $\alpha, \beta$ with MacKay's algorithm

$$\alpha \leftarrow \frac{\gamma}{\boldsymbol{m}^{\mathrm{T}}\boldsymbol{m}}, \beta \leftarrow \frac{N - \gamma}{\|\mathbf{F}\boldsymbol{m} - \boldsymbol{t}\|^2}, \tag{4}$$

model evidence can be maximized with $\alpha, \beta$ optimized simultaneously.

Linear probe is usually used as a method of transfer learning, especially when the pre-trained backbone is large (Radford et al., 2021) as the retraining of large models is sometimes prohibitive. In such a setting, pre-trained backbones are used to extract representations, and a fully connected layer is used to predict results based on the representations. This process is a linear transform, and the model evidence can be calculated analytically.

Therefore, to assess and compare the transferability of different models on target task $\mathcal{D}$, for each pre-trained backbone $\mathcal{M}_i$, evidence $\mathcal{L}_i(\alpha, \beta)$ can be calculated and maximized with MacKays's algorithm, resulting in the logarithm of maximum evidence $\mathcal{L}_i(\alpha^*, \beta^*)$. Eventually, transferability can be compared in terms of $\mathcal{L}(\alpha^*, \beta^*)$, and candidate models can be ranked.

### 3.2 Adapting maximum evidence method to SNN with averaged features

Architectural differences between SNNs and ANNs impede the direct application of existing feature-based transferability assessment methods for ANNs to SNNs. SNN uses multiple time steps of forward pass to generate results. Therefore, the feature map extracted by an SNN backbone is a two-dimensional matrix instead of a single-dimensional vector. In addition, SNN models generate predictions based on the feature map with a non-linear transform in classification tasks.

We consider the most used leaky-integral and fire (LIF) spiking neuron model. One layer of LIF neurons in the SNN is given by

$$\begin{cases} \tau \dfrac{\mathrm{d}\boldsymbol{u}^n(t)}{\mathrm{d}t} = -\boldsymbol{u}^n(t) + \boldsymbol{\Theta}^n \boldsymbol{o}^{n-1}(t), \\ o_i^n(t) = 1, \text{if } u_i^n(t) \geq u_{th}, \\ o_i^n(t) = 0, \text{if } u_i^n(t) < u_{th}, \end{cases} \tag{5}$$

where $n$ is the layer number of the multilayer SNN, $t$ is time and $\tau > 0$ is a time constant. Each neuron in the last layer is depicted by its membrane potential $u \in \mathbb{R}$ and output spike $o \in \{0, 1\}$. The membrane potential $u_i^n(t)$ will be reset to $u_0$ (usually $u_0 = 0$) if $o_i^n(t) = 1$. The spiking frequency is usually used as the prediction of logit of the target category. This nonlinear transform makes existing ANN transferability assessment methods incompatible with SNNs.

We propose to use features averaged through time steps for the evaluation of the transferability of SNN PTMs. Consider a pre-trained SNN model with $T$ time steps, for data $(\boldsymbol{x}, t) \in \mathcal{D}$, the feature map extracted by the model is $\boldsymbol{f}(t) \in \mathbb{R}^D, t = 1, \cdots, T$. We denote averaged feature as $\bar{\boldsymbol{f}} = \frac{1}{T} \sum_{t=1}^{T} \boldsymbol{f}(t) \in \mathbb{R}^D$. To get the output prediction $\hat{t}$, the feature map $\boldsymbol{f}(t)$ is processed by equation 5 as

$$u(t) = \mathrm{e}^{-\frac{\Delta t}{\tau}} u(t-1) \left(1 - o(t-1)\right) + \boldsymbol{\theta}^{\mathrm{T}} \boldsymbol{f}(t). \tag{6}$$

$$o(t) = s(u(t) - u_{th}), \tag{7}$$

$$\hat{t} = \frac{1}{T} \sum_{t=1}^{T} o(t), \tag{8}$$

where $\mathrm{e}^{-\frac{\Delta t}{\tau}}$ denotes the decay of membrane potential in a time step $\Delta t$, time constant $\tau$ describes the speed to decay, $s(\cdot)$ is the function to describe the spiking behavior of a spiking neuron, giving 1 when input is non-negative and 0 otherwise, and $u_{th}$ is the threshold to spike.

In classification tasks where the target $t \in \{0, 1\}$, each predicted result $\hat{t}$ should tend towards its target $t$. Suppose that $t = \hat{t} = 1$, then $o(t) = 1$ for all $t = 1, \cdots, T$. Therefore, the membrane potential in each time step is a linear transform of the feature of this time step as

$$u(t) = \boldsymbol{\theta}^{\mathrm{T}} \boldsymbol{f}(t). \tag{9}$$

From equation 7, it is clear that $\boldsymbol{\theta}^{\mathrm{T}} \boldsymbol{f}^{(t)} = u^{(t)} \geq u_{th}, \forall t = 1, \dots, T$. Therefore, we can get that

$$\boldsymbol{\theta}^{\mathrm{T}} \bar{\boldsymbol{f}} = \boldsymbol{\theta}^{\mathrm{T}} \left( \frac{1}{T} \sum_{t=1}^{T} \boldsymbol{f}(t) \right) = \frac{1}{T} \sum_{t=1}^{T} \boldsymbol{\theta}^{\mathrm{T}} \boldsymbol{f}(t) \geq \frac{1}{T} \sum_{t=1}^{T} u_{th} = u_{th}. \tag{10}$$

Conversely, when $t = \hat{t} = 0$, we have $\boldsymbol{\theta}^{\mathrm{T}} \bar{\boldsymbol{f}} \leq u_{th}$. To keep the main text concise, the derivation for this case is provided in Appendix A.2.3. This shows that the SNN transform of a single layer can be approximated by a linear classification problem, and then transformed into a linear problem on average features.

### 3.3 Accelerating convergence with approximated maximum evidence

MacKay's algorithm iteratively updates the regularization coefficients and maximizes the model evidence. However, the convergence of MacKay's algorithm is not guaranteed (You et al., 2022), and the iteration

number might be large in some cases. To reduce the time consumption and thus further improve the efficiency of model hub scheduling, here we propose an alternative method that avoids the iteration while giving an approximated result of maximum model evidence with MacKay's algorithm (hereinafter referred to as MacKay's method). We name our proposed method *approximated maximum model evidence* method (hereinafter referred to as *the approximated method*).

The starting point of designing the approximated version is to make sure that our approximated results share the same properties as MacKay's method. To achieve this, we state three properties of MacKay's method. These properties describe the invariance of MacKay's method with some operation on the input feature map. We design our approximated method adhering to these properties.

**Property 1** (invariance with stacking)**.** *Consider input data* $\mathbf{F} \in \mathbb{R}^{N \times D}$ *and targets* $\boldsymbol{t} \in \mathbb{R}^N$. *Suppose that MacKay's algorithm converges to* $\alpha^*$ *and* $\beta^*$ *and the logarithm of maximized model evidence is* $\mathcal{L}_1^*$, *then when input data is stacked* $q$ *times as* $\left[\mathbf{F}, \ldots, \mathbf{F}\right] \in \mathbb{R}^{N \times qD}$, *MacKay's method converges to* $q\alpha^*$ *and* $\beta^*$, *and the logarithm of maximized model evidence* $\mathcal{L}_2^*$ *equals exactly to* $\mathcal{L}_1^*$.

**Property 2** (invariance with padding zero)**.** *Consider input data* $\mathbf{F} \in \mathbb{R}^{N \times D}$ *and targets* $\boldsymbol{t} \in \mathbb{R}^N$. *Suppose that MacKay's algorithm converges to* $\alpha^*$ *and* $\beta^*$ *and the logarithm of maximized model evidence is* $\mathcal{L}_1^*$, *then when input data is padded with zeros as* $\left[\mathbf{F}, \mathbf{0}\right] \in \mathbb{R}^{N \times qD}$, *MacKay's method converges to* $\alpha^*$ *and* $\beta^*$, *and the logarithm of maximized model evidence* $\mathcal{L}_2^*$ *equals exactly to* $\mathcal{L}_1^*$.

**Property 3** (invariance with scalar multiplication)**.** *Consider input data* $\mathbf{F} \in \mathbb{R}^{N \times D}$ *and targets* $\boldsymbol{t} \in \mathbb{R}^N$. *Suppose that MacKay's algorithm converges to* $\alpha^*$ *and* $\beta^*$ *and the logarithm of maximized model evidence is* $\mathcal{L}_1^*$, *then when input data is multiplied by a scalar* $q > 0$ *as* $q\mathbf{F} \in \mathbb{R}^{N \times D}$, *MacKay's method converges to* $q^2\alpha^*$ *and* $\beta^*$, *and the logarithm of maximized model evidence* $\mathcal{L}_2^*$ *equals exactly to* $\mathcal{L}_1^*$.

The proof of property 1 and property 2 can be found in reference You et al. (2022). The proof of property 3 is given in appendix A.2.1.

The approximated maximum model evidence method is given below.

Consider input feature $\mathbf{F} \in \mathbb{R}^{N \times D}$ and labels $\boldsymbol{t} \in \mathbb{R}^N$. Let

$$\lambda \leftarrow \frac{\|\mathbf{F}\|_2^2}{N}, \tag{11}$$

$$\boldsymbol{m} \leftarrow \left(\lambda \mathbf{I} + \mathbf{F}^{\mathrm{T}} \mathbf{F}\right)^{-1} \mathbf{F}^{\mathrm{T}} \boldsymbol{t}, \tag{12}$$

$$\beta_0 \leftarrow \frac{N}{\|\mathbf{F}\boldsymbol{m} - \mathbf{t}\|^2 + \lambda \|\boldsymbol{m}\|^2}, \tag{13}$$

$$\alpha_0 \leftarrow \lambda \beta_0. \tag{14}$$

Then the approximate model evidence can be calculated by equation 3 with $\alpha_0$ and $\beta_0$.

**Theorem 1.** *The model evidence calculated by our method satisfies properties 1, 2, and 3.*

*Proof.* See appendix A.2.2. □

The experimental results in section 4.4 show that our method gives similar results to MacKay's method when the number of samples $N$ is larger than the dimension of features $D$. Therefore, the approximated result can serve as an approximated maximum model evidence under such circumstances. We have summarized the procedure of MEAF with MacKay's method and our approximated method in Algorithm 1.

## 4 Results

### 4.1 Experimental setup

**Datasets** The experiment is conducted on both neuromorphic datasets and static datasets. Neuromorphic datasets are vision datasets recorded with event cameras (Gallego et al., 2020), which are the commonly

---

**Algorithm 1** Algorithm of maximum evidence method with averaged feature

---

1: **Input:** Feature map of an SNN PTM on a target dataset $\{\mathbf{F}(t) \in \mathbb{R}^{N \times D}\}_{t=0}^{T-1}$, labels $\boldsymbol{t} \in \mathbb{R}^N$
2: **Output:** Score of transferability $\mathcal{L}$
3: **Step 1:** calculate averaged feature through time $\bar{\mathbf{F}} = \sum_{t=0}^{T-1} \mathbf{F}(t)$
4: **Step 2 (MacKay's method):**
5: $\alpha = 1,\ \beta = 1$
6: **repeat**
7: $\quad \lambda \leftarrow \frac{\alpha}{\beta}$
8: $\quad \mathbf{A} \leftarrow \alpha \mathbf{I} + \beta \bar{\mathbf{F}}^{\mathrm{T}} \bar{\mathbf{F}}$
9: $\quad \boldsymbol{m} \leftarrow \mathbf{A}^{-1} \bar{\mathbf{F}}^{\mathrm{T}} \boldsymbol{t}$
10: $\quad \alpha \leftarrow \frac{\gamma}{\boldsymbol{m}^{\mathrm{T}} \boldsymbol{m}},\ \beta \leftarrow \frac{N-\gamma}{||\bar{\mathbf{F}}\boldsymbol{m}-\mathbf{t}||^2}$
11: $\quad \lambda' \leftarrow \frac{\alpha}{\beta}$
12: **until** $|\lambda - \lambda'| < \epsilon$
13: **Step 2 (approximated method):**
14: $\lambda \leftarrow \frac{||\mathbf{F}||^2}{N}$
15: $\boldsymbol{m} \leftarrow \left(\lambda \mathbf{I} + \bar{\mathbf{F}}^{\mathrm{T}} \bar{\mathbf{F}}\right)^{-1} \bar{\mathbf{F}}^{\mathrm{T}} \boldsymbol{t}$
16: $\beta \leftarrow \frac{N}{||\bar{\mathbf{F}}\boldsymbol{m}-\mathbf{t}||^2 + \lambda ||\boldsymbol{m}||^2}$
17: $\alpha \leftarrow \lambda \beta$
18: $\mathbf{A} \leftarrow \alpha \mathbf{I} + \beta \bar{\mathbf{F}}^{\mathrm{T}} \bar{\mathbf{F}}$
19: **Step 3:** calculate score of transferability $\mathcal{L}$ by equation 3 with $\mathbf{A}, \boldsymbol{m}, \alpha, \beta$

---

used benchmark for SNNs (He et al., 2020). As SNNs can also apply to conventional frame-based images (static datasets), we also validate our method on these datasets. For experiments on neuromorphic datasets, we use ES-ImageNet dataset (Lin et al., 2021) for pre-training and DVS128 Gesture (DVS128), CIFAR10-DVS (C10-DVS), N-Caltech101 (N-C101) and N-MNIST for fine-tuning, and for that on static datasets, the SNN models are pre-trained on ImageNet (Deng et al., 2009) and fine-tuned on CIFAR10 (C10), CIFAR100 (C100), Caltech101 (C101) and MNIST.

**Pre-train SNN models** We pre-train SNN models as the foundation of our experiment. We use spiking version of vanilla ResNet, spiking version of multi-layer perception (MLP), spike-element-wise (SEW) ResNet (Fang et al., 2021) and attention spiking neural networks (Yao et al., 2023) as candidate PTMs. Ten different kinds of SNN models are used as different kinds of PTMs on neuromorphic datasets, and eight different kinds of SNN models are used as PTMs on static datasets. The details of the PTMs are given in appendix A.1. For experiments on neuromorphic datasets, models are pre-trained with Adam optimizer (Kingma & Ba, 2014) with momentum 0.9 and 0.999 for 10 epochs with 8 RTX3090 GPUs. We use a step learning rate scheduler with an initial learning rate of 0.01, step size 3, and $\gamma = 0.3$. For experiments on static datasets, we use PTMs from Fang et al. (2021).

**Fine-tune pre-trained models** The fine-tuning of pre-trained SNN models are as the settings in logME (You et al., 2021). The parameters of the pre-trained neural network backbone are frozen during fine-tuning, and a spiking FC layer is re-trained as the classifier of the PTMs. On neuromorphic datasets, each model is re-trained for 200 epochs. The hyper-parameters, learning rate, and weight decay, are determined by grid-searching from $1e-1$ to $1e-3$ and $1e-5$ to $1e-8$ respectively. On static datasets, each model is re-trained for 100 epochs. The same hyper-parameters are determined by grid-searching from $1e-1$ to $1e-3$ and $1e-5$ to $1e-7$ respectively. The test set accuracy of the best model on the validation set is used as the ground truth of transferability.

**Assessment of transferability** For each PTM, the transferability is assessed with MEAF, LEEP, and NCE [1]. We calculate the logarithm of maximum model evidence on the train set. Note that ranking models with the maximum model evidence does not require the involvement of the validation or test set. This is also

---

[1] Code is available at https://github.com/haohq19/meaf-snn.

an advantage of the assessment method over the fine-tuning methods. Same as You et al. (2021), the Kendall correlation coefficient is used as a metric to evaluate the validity of our assessed score of transferability. Given $N$ candidate models, $\{T_i, 1 \leq i \leq N\}$ is the list of their ground truth transferability and $\{S_i, 1 \leq i, \leq N\}$ is the list of predicted scores. Kendall's coefficient $\tau$ gives a quantitative illustration of how the ranking of $S_i$ can stand for the ranking of $T_i$ that when $S_i > S_j$, then the probability that $T_i > T_j$ is $\frac{\tau+1}{2}$ (Fagin et al., 2003). We also provide Pearson correlation coefficients between the test accuracy and predicted scores, which is used in LEEP (Nguyen et al., 2020) and NCE (Tran et al., 2019) as the metric. This metric could better illustrate how the predicted scores are linearly correlated with the ground truth transferability.

## 4.2 Transferability assessment on neuromorphic datasets

We demonstrate the effectiveness of the maximum evidence method with averaged features on neuromorphic datasets. The results of Kendall correlation coefficients are shown in Figure 1. It is shown that our MEAF method gives the best prediction result on all datasets, compared to NCE and LEEP. The Kendall coefficients are above 0.78 on all datasets, meaning that the predicted scores can represent the ranking of real transferability with a probability of more than 89%. It is shown that LEEP also gives effective predictions on SNNs but with a relatively lower Kendall coefficient. Note that on N-MNIST, NCE gives predictions with correlation coefficients nearly equal to zero.

The Pearson correlation coefficients are shown in Table 1, which shows that MEAF gives predicted scores more linearly correlated with the ground truth transferability. Neuromorphic datasets have an additional temporal dimension, which results in a more complicated feature map extracted by the SNNs (He et al., 2020; Samadzadeh et al., 2023; Zhang et al., 2023). Therefore, making assessments of transferability based on the feature map could leverage more information than based on the outputs, which may downgrade the performance LEEP and NCE on such datasets.

Table 1: Pearson correlation coefficients of NCE, LEEP and MEAF on neuromorphic datasets

| DATASET | C10-DVS | DVS128 | N-MNIST | N-C101 |
|---------|---------|--------|---------|--------|
| NCE     | 0.67    | 0.08   | 0.11    | 0.63   |
| LEEP    | 0.76    | 0.31   | 0.24    | 0.57   |
| MEAF    | **0.99**| **0.91**| **0.97**| **0.96**|

## 4.3 Transferability assessment on static datasets

Our experimental results of Kendall correlation coefficients demonstrate the effectiveness of the MEAF method on static datasets, as illustrated in Figure 2. The Kendall coefficients shows that the MEAF approach achieves comparable performance with NCE and LEEP on CIFAR10 and CIFAR100 datasets. Notably, MEAF outperforming both NCE and LEEP by a margin of 0.5 on the MNIST dataset. While MEAF shows comparable performance to NCE on the Caltech101 dataset, it exhibits a marginal difference of 0.14 compared to LEEP. When compared with the results obtained from neuromorphic datasets, the Kendall coefficients of MEAF in this experiment show relatively lower values. However, it is noteworthy that all coefficients remain above 0.5 across all datasets, indicating that the predicted scores can effectively represent the ranking of real transferability with a probability of more than 0.75. We suspect that this is because the transferabilities (represented by the test set accuracy) of the PTMs are much closer in this experiment. The Pearson coefficients are given in Table 2. MEAF shows better linear correlation with the ground truth transferability compared with LEEP and NCE on CIFAR10, CIFAR100 and MNIST datasets, and achieves the second performance on the Caltech101 dataset behind LEEP. Overall, the experiment results on the static datasets show that MEAF gives comparable results as LEEP and NCE. Static datasets do not have a temporal dimension, so they could not leverage the temporal dynamics of SNNs. SNNs act like ANNs on the static datasets, which could explain that LEEP and NCE perform well here.

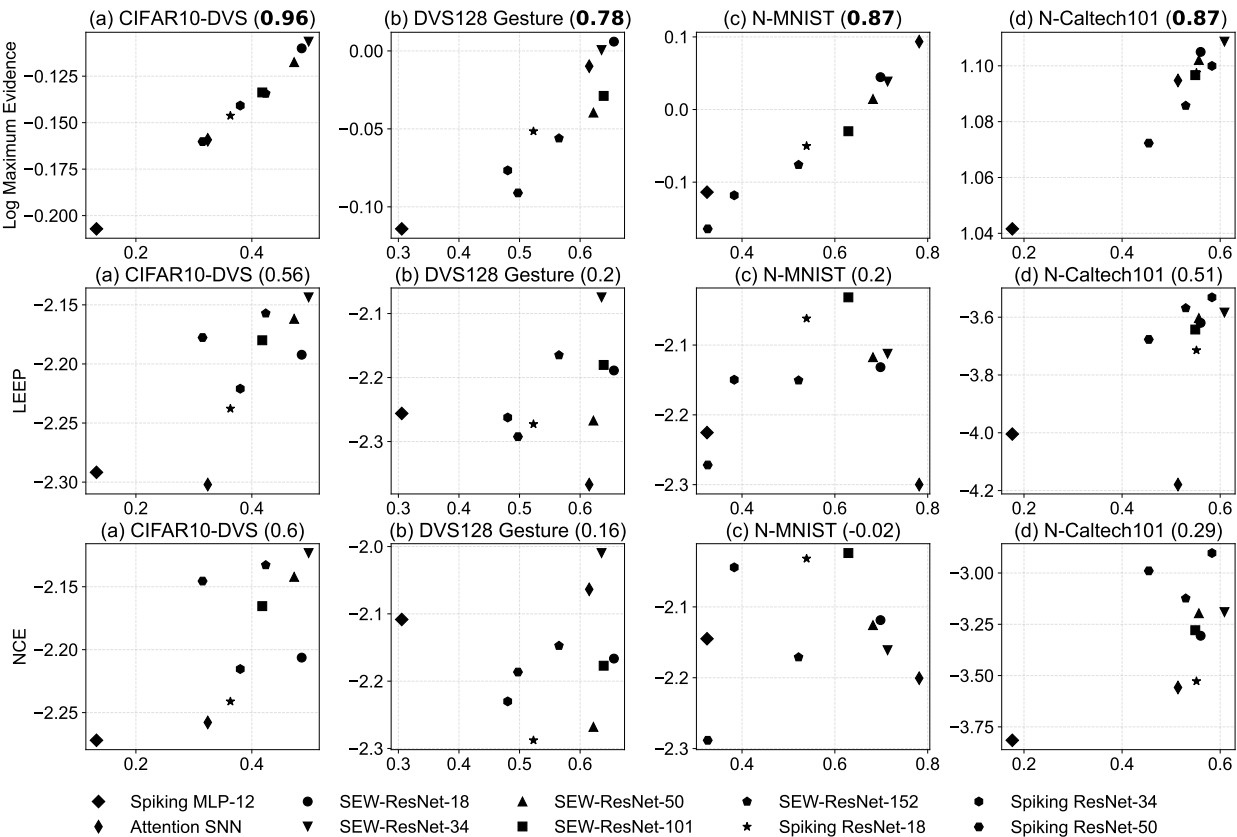

Figure 1: Result of **assessed transferability scores (y-axis) vs. test set accuracy (x-axis)** on 4 neuromorphic datasets. The first row gives the result of the MEAF. The second and third row shows the results of LEEP and NCE respectively. Kendall correlation coefficients are labeled in the title of each subplot after the dataset name.

Table 2: Pearson correlation coefficients of NCE, LEEP and MEAF on static datasets

| DATASET | C10 | C100 | C101 | MNIST |
|---|---|---|---|---|
| NCE | 0.89 | 0.87 | 0.62 | 0.37 |
| LEEP | 0.90 | 0.90 | **0.80** | 0.38 |
| MEAF | **0.95** | **0.91** | 0.64 | **0.83** |

## 4.4 Validation of approximated maximum evidence

We first show that the approximated maximum evidence method gives consistent results with MacKay's method when the number of samples is large. In this experiment, we generate toy data for linear regression and add Gaussian noise to the data with different SNRs from -10 dB to 10 dB. The dimension of the features (D) is fixed at 100 and the number of samples (N) increases from 100 to 3,000. The results are illustrated in Figure 3, showing that the approximated maximum evidence converges to MacKay's results as N tends to infinity.

We then validate that the approximated maximum evidence is effective in model hub scheduling. Figure 4 shows that with the approximated maximum evidence, the Kendall coefficients are above 0.85 on CIFAR10-DVS, N-MNIST, and N-Caltech101 datasets, which is not lower than the results of MacKay's method. On

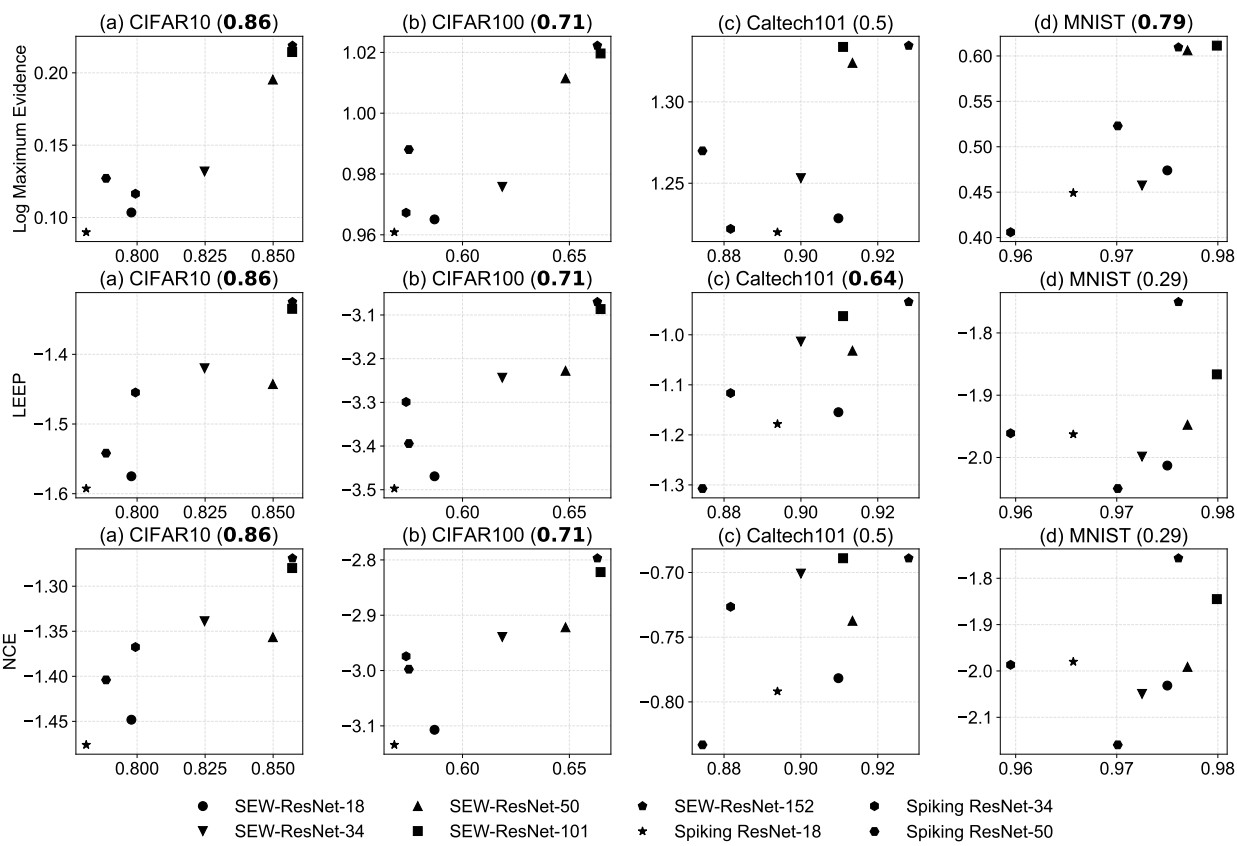

Figure 2: Result of **assessed transferability scores (y-axis) vs. test set accuracy (x-axis)** on 4 static datasets. The first row gives the result of the MEAF. The second and third row shows the results of LEEP and NCE respectively. Kendall correlation coefficients are labeled in the title of each subplot after the dataset name.

the DVS128 Gesture dataset, the Kendall coefficient is 0.56, which is still higher than LEEP (0.2) and NCE (0.16). We believe that the drop of coefficient on DVS128 Gesture dataset is because the DVS128 Gesture is a small dataset, with only 1176 samples in the training set (Amir et al., 2017). This contradicts our assumption of the number of samples being much greater than the dimension of features. As shown in Figure 3, when $N \approx D$, the result of approximated method differs a lot from the results of MacKay's method. The results on static datasets (Figure 5) give the Kendall coefficient not lower than the results by MacKay's method on CIFAR10 and CIFAR100 datasets. The result on the Caltech101 dataset gives an Kendall coefficient of 0.43, which is 0.07 lower than the result of MacKays's method (0.5). For the MNIST dataset, the Kendall coefficient is 0.64, 0.15 lower than the result of MacKay's method (0.79), but still comparable to the results given by LEEP (0.29) and NCE (0.29). We believe this might be due to the fact that the test set accuracy of the PTMs on MNIST is overly concentrated, with values distributed between approximately 0.96 and 0.98. As a result, the transferability assessment results are also very similar, making the impact of errors more significant. This might explain the noticeable decline in the correlation coefficient on this dataset. The Pearson correlation coefficients are given in Table 3. It is shown that the approximated method gives similar results with MacKay's method on most of the datasets.

Here we give the average number of iterations required for MacKay's method in Table 4. The stopping criterion for the convergence of MacKay's algorithm is set such that the relative error between two successive $\alpha/\beta$ is less than 0.01. This is a relatively lenient stopping criterion; employing a stricter criterion would result in a higher number of iterations. Note that the approximated method does not need an iteration

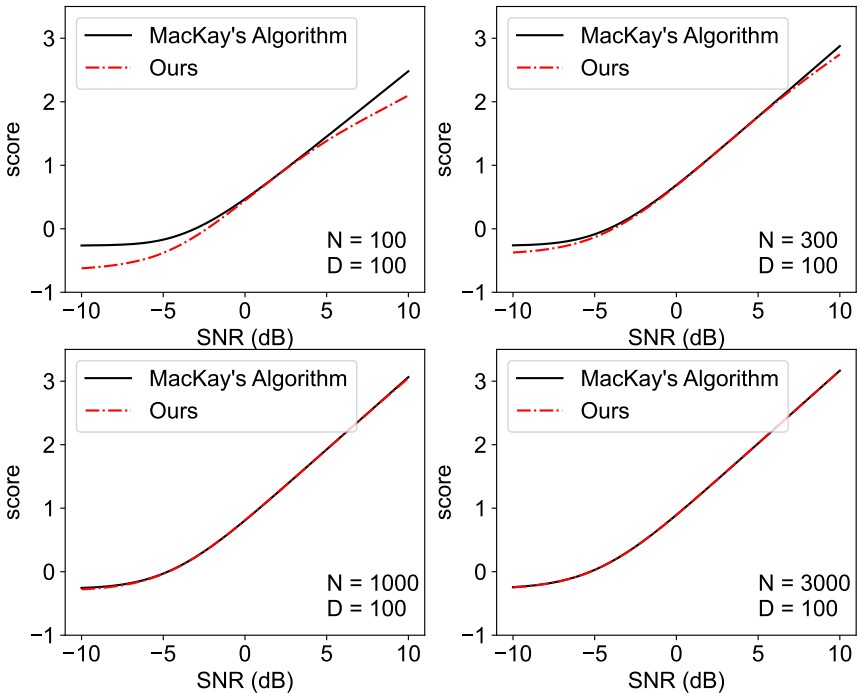

Figure 3: Consistency between the results of MacKay's algorithm and our approximated result on toy data. Dimension of feature (D) is fixed as 100 and the number of samples (N) is 100, 3,000, 1,000, and 3,000 from (a) to (d). The abscissa represents the SNR of the toy data, and the ordinate represents the scores of transferability given by MacKay's method and our approximated method.

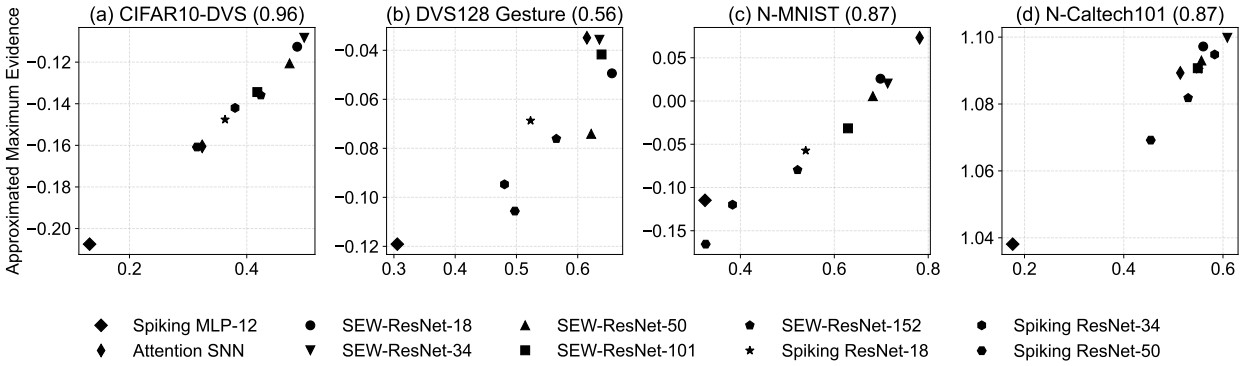

Figure 4: Result of **assessed transferability scores (y-axis) vs. test set accuracy (x-axis)** on 4 neuromorphic datasets. The abscissa represents the test set accuracy on the corresponding dataset, and the ordinate represents the scores of transferability given by the approximated maximum evidence method. Kendall correlation coefficients are labeled in the title of each subplot after the dataset name.

procedure. Therefore, it could accelerate the assessment of transferability from about 1× to 7× on these datasets.

We summarized the Kendall coefficients of NCE, LEEP, MEAF (with MacKay's method), MEAF (with Approximated method) on all the datasets in Table 5.

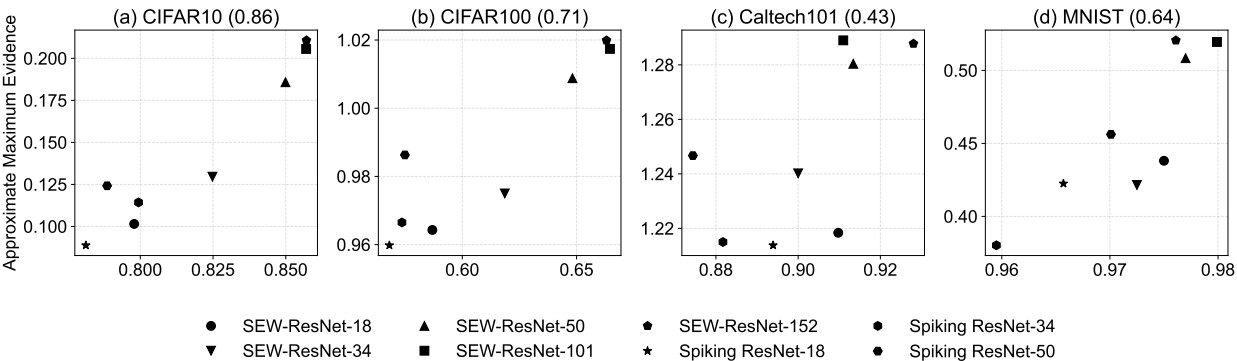

Figure 5: Result of **assessed transferability scores (y-axis) vs. test set accuracy (x-axis)** on 4 static datasets. The abscissa represents the test set accuracy on the corresponding dataset, and the ordinate represents the scores of transferability given by the approximated maximum evidence method. Kendall correlation coefficients are labeled in the title of each subplot after the dataset name.

Table 3: Pearson coefficients of MEAF with MacKay's method and approximated method

| Dataset | C10-DVS | DVS128 | N-MNIST | N-C101 | C10 | C100 | C101 | MNIST |
|---|---|---|---|---|---|---|---|---|
| MEAF (+MacKay) | **0.99** | **0.91** | **0.97** | 0.96 | 0.95 | **0.91** | 0.64 | 0.83 |
| MEAF (+Appox) | **0.99** | 0.87 | **0.97** | **0.98** | **0.96** | **0.91** | 0.63 | **0.86** |

## 5 Discussion

This paper introduces a novel approach to assessing the transferability of SNN PTMs. We highlight the challenges of directly applying the existing Bayesian method to SNN transferability assessment and propose the MEAF method to address these issues. Additionally, an approximated maximum evidence method is introduced to reduce the computational time of model hub scheduling. We hope this work can help SNN PTMs be more widely and efficiently used.

Our work is limited in the diversity of SNN PTMs. There are currently no large-scale SNN model hubs, and our computational resources for pre-training are restricted. A more extensive collection of SNN PTMs would enhance the accuracy and generalizability of our conclusions.

### Acknowledgments

This work was supported by the National Key Research and Development Program of China (Grant No. 2021ZD0200300).

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

Table 4: Iteration number used for MacKay's method

| MODEL | C10-DVS | DVS128 | N-MNIST | N-C101 | C10 | C100 | C101 | MNIST |
|---|---|---|---|---|---|---|---|---|
| Spiking MLP-12 | 4.9 | 6.8 | 3.0 | 11.9 | – | – | – | – |
| Attention SNN | 6.6 | 8.3 | 3.7 | 8.8 | – | – | – | – |
| SEW ResNet-18 | 6.0 | 5.2 | 2.3 | 7.9 | 2.0 | 3.2 | 4.3 | 1.1 |
| SEW ResNet-34 | 5.6 | 5.2 | 3.3 | 7.6 | 2.0 | 3.0 | 4.0 | 1.4 |
| SEW ResNet-50 | 4.5 | 6.3 | 4.0 | 7.1 | 3.1 | 4.4 | 6.5 | 2.0 |
| SEW ResNet-101 | 3.6 | 6.6 | 3.5 | 6.3 | 3.2 | 4.3 | 6.3 | 2.1 |
| SEW ResNet-152 | 4.1 | 6.8 | 3.7 | 5.3 | 3.1 | 4.2 | 6.3 | 2.0 |
| Spiking ResNet-18 | 6.0 | 5.3 | 3.1 | 8.0 | 2.1 | 3.4 | 4.6 | 1.1 |
| Spiking ResNet-34 | 5.0 | 5.3 | 2.9 | 7.0 | 2.1 | 3.4 | 4.8 | 0.7 |
| Spiking ResNet-50 | 3.5 | 5.9 | 2.9 | 6.2 | 3.2 | 5.0 | 7.2 | 2.3 |
| Average | 5.0 | 6.2 | 3.2 | 7.6 | 2.6 | 3.9 | 5.5 | 1.6 |

Table 5: Kendall coefficients of NCE, LEEP and MEAF

| DATASET | C10-DVS | DVS128 | N-MNIST | N-C101 | C10 | C100 | C101 | MNIST | AVERAGE |
|---|---|---|---|---|---|---|---|---|---|
| NCE | 0.60 | 0.16 | -0.02 | 0.29 | **0.86** | **0.71** | 0.50 | 0.29 | 0.42 |
| LEEP | 0.56 | 0.20 | 0.20 | 0.51 | **0.86** | **0.71** | **0.64** | 0.29 | 0.50 |
| MEAF(MacKay) | **0.96** | **0.78** | **0.87** | **0.87** | **0.86** | **0.71** | 0.5 | **0.79** | **0.79** |
| MEAF(Approx) | **0.96** | 0.56 | **0.87** | **0.87** | **0.86** | **0.71** | 0.43 | 0.64 | 0.74 |

Zhenshan Bing, Zhuangyi Jiang, Long Cheng, Caixia Cai, Kai Huang, and Alois Knoll. End to end learning of a multi-layered SNN based on R-STDP for a target tracking snake-like robot. In *2019 International Conference on Robotics and Automation (ICRA)*, pp. 9645–9651. IEEE, 2019.

Tom Brown, Benjamin Mann, Nick Ryder, Melanie Subbiah, Jared D Kaplan, Prafulla Dhariwal, Arvind Neelakantan, Pranav Shyam, Girish Sastry, Amanda Askell, et al. Language models are few-shot learners. *Advances in neural information processing systems*, 33:1877–1901, 2020.

Manon Dampfhoffer, Thomas Mesquida, Alexandre Valentian, and Lorena Anghel. Are snns really more energy-efficient than anns? an in-depth hardware-aware study. *IEEE Transactions on Emerging Topics in Computational Intelligence*, 2022.

Erik Daxberger, Agustinus Kristiadi, Alexander Immer, Runa Eschenhagen, Matthias Bauer, and Philipp Hennig. Laplace redux-effortless Bayesian deep learning. *Advances in Neural Information Processing Systems*, 34:20089–20103, 2021.

Guillaume Debat, Tushar Chauhan, Benoit R Cottereau, Timothée Masquelier, Michel Paindavoine, and Robin Baures. Event-based trajectory prediction using spiking neural networks. *Frontiers in computational neuroscience*, 15:658764, 2021.

Jia Deng, Wei Dong, Richard Socher, Li-Jia Li, Kai Li, and Li Fei-Fei. Imagenet: A large-scale hierarchical image database. In *2009 IEEE conference on computer vision and pattern recognition*, pp. 248–255. Ieee, 2009.

Jacob Devlin, Ming-Wei Chang, Kenton Lee, and Kristina Toutanova. Bert: Pre-training of deep bidirectional transformers for language understanding. *arXiv preprint arXiv:1810.04805*, 2018.

Zhengxiao Du, Yujie Qian, Xiao Liu, Ming Ding, Jiezhong Qiu, Zhilin Yang, and Jie Tang. Glm: General language model pretraining with autoregressive blank infilling. In *Proceedings of the 60th Annual Meeting of the Association for Computational Linguistics (Volume 1: Long Papers)*, pp. 320–335, 2022.

Ronald Fagin, Ravi Kumar, and Dakshinamurthi Sivakumar. Comparing top k lists. *SIAM Journal on discrete mathematics*, 17(1):134–160, 2003.

Wei Fang, Zhaofei Yu, Yanqi Chen, Tiejun Huang, Timothée Masquelier, and Yonghong Tian. Deep residual learning in spiking neural networks. *Advances in Neural Information Processing Systems*, 34:21056–21069, 2021.

Guillermo Gallego, Tobi Delbrück, Garrick Orchard, Chiara Bartolozzi, Brian Taba, Andrea Censi, Stefan Leutenegger, Andrew J Davison, Jörg Conradt, Kostas Daniilidis, et al. Event-based vision: A survey. *IEEE transactions on pattern analysis and machine intelligence*, 44(1):154–180, 2020.

Stephen F Gull. Developments in maximum entropy data analysis. In *Maximum Entropy and Bayesian Methods: Cambridge, England, 1988*, pp. 53–71. Springer, 1989.

Jesse Hagenaars, Federico Paredes-Vallés, and Guido De Croon. Self-supervised learning of event-based optical flow with spiking neural networks. *Advances in Neural Information Processing Systems*, 34:7167–7179, 2021.

Kaiming He, Ross Girshick, and Piotr Dollár. Rethinking imagenet pre-training. In *Proceedings of the IEEE/CVF International Conference on Computer Vision*, pp. 4918–4927, 2019.

Weihua He, YuJie Wu, Lei Deng, Guoqi Li, Haoyu Wang, Yang Tian, Wei Ding, Wenhui Wang, and Yuan Xie. Comparing SNNs and RNNs on neuromorphic vision datasets: Similarities and differences. *Neural Networks*, 132:108–120, 2020.

Weihua He, Junwen Zhu, Yongxiang Feng, Fei Liang, Kaichao You, Huichao Chai, Zhipeng Sui, Haiqing Hao, Guoqi Li, Jingjing Zhao, et al. Neuromorphic-enabled video-activated cell sorting. *Nature Communications*, 15(1):10792, 2024.

Alexander Immer, Matthias Bauer, Vincent Fortuin, Gunnar Rätsch, and Mohammad Emtiyaz Khan. Scalable marginal likelihood estimation for model selection in deep learning. In *International Conference on Machine Learning*, pp. 4563–4573. PMLR, 2021.

Chao Jia, Yinfei Yang, Ye Xia, Yi-Ting Chen, Zarana Parekh, Hieu Pham, Quoc Le, Yun-Hsuan Sung, Zhen Li, and Tom Duerig. Scaling up visual and vision-language representation learning with noisy text supervision. In *International conference on machine learning*, pp. 4904–4916. PMLR, 2021.

Junguang Jiang, Yang Shu, Jianmin Wang, and Mingsheng Long. Transferability in deep learning: A survey. *arXiv preprint arXiv:2201.05867*, 2022.

Yong-Deok Kim, Taewoong Jang, Bohyung Han, and Seungjin Choi. Learning to select pre-trained deep representations with Bayesian evidence framework. In *Proceedings of the IEEE Conference on Computer Vision and Pattern Recognition*, pp. 5318–5326, 2016.

Diederik P Kingma and Jimmy Ba. Adam: A method for stochastic optimization. *arXiv preprint arXiv:1412.6980*, 2014.

Kevin H Knuth, Michael Habeck, Nabin K Malakar, Asim M Mubeen, and Ben Placek. Bayesian evidence and model selection. *Digital Signal Processing*, 47:50–67, 2015.

Simon Kornblith, Jonathon Shlens, and Quoc V Le. Do better imagenet models transfer better? In *Proceedings of the IEEE/CVF conference on computer vision and pattern recognition*, pp. 2661–2671, 2019.

Frederik Kunstner, Philipp Hennig, and Lukas Balles. Limitations of the empirical Fisher approximation for natural gradient descent. *Advances in neural information processing systems*, 32, 2019.

Chankyu Lee, Priyadarshini Panda, Gopalakrishnan Srinivasan, and Kaushik Roy. Training deep spiking convolutional neural networks with STDP-based unsupervised pre-training followed by supervised fine-tuning. *Frontiers in neuroscience*, 12:435, 2018.

Yihan Lin, Wei Ding, Shaohua Qiang, Lei Deng, and Guoqi Li. Es-imagenet: A million event-stream classification dataset for spiking neural networks. *Frontiers in neuroscience*, 15:1546, 2021.

Yihan Lin, Yifan Hu, Shijie Ma, Dongjie Yu, and Guoqi Li. Rethinking pretraining as a bridge from ANNs to SNNs. *IEEE Transactions on Neural Networks and Learning Systems*, 2022.

Wolfgang Maass. Networks of spiking neurons: the third generation of neural network models. *Neural networks*, 10(9):1659–1671, 1997.

David JC MacKay. Bayesian interpolation. *Neural computation*, 4(3):415–447, 1992a.

David JC MacKay. The evidence framework applied to classification networks. *Neural computation*, 4(5): 720–736, 1992b.

David JC MacKay. A practical bayesian framework for backpropagation networks. *Neural computation*, 4 (3):448–472, 1992c.

David JC MacKay. Comparison of approximate methods for handling hyperparameters. *Neural computation*, 11(5):1035–1068, 1999.

TorchVision maintainers and contributors. Torchvision: Pytorch's computer vision library. `https://github.com/pytorch/vision`, 2016.

Yishay Mansour, Mehryar Mohri, and Afshin Rostamizadeh. Domain adaptation: Learning bounds and algorithms. *arXiv preprint arXiv:0902.3430*, 2009.

Cuong Nguyen, Tal Hassner, Matthias Seeger, and Cedric Archambeau. Leep: A new measure to evaluate transferability of learned representations. In *International Conference on Machine Learning*, pp. 7294–7305. PMLR, 2020.

Jing Pei, Lei Deng, Sen Song, Mingguo Zhao, Youhui Zhang, Shuang Wu, Guanrui Wang, Zhe Zou, Zhenzhi Wu, Wei He, et al. Towards artificial general intelligence with hybrid tianjic chip architecture. *Nature*, 572(7767):106–111, 2019.

Alec Radford, Jeffrey Wu, Rewon Child, David Luan, Dario Amodei, Ilya Sutskever, et al. Language models are unsupervised multitask learners. *OpenAI blog*, 1(8):9, 2019.

Alec Radford, Jong Wook Kim, Chris Hallacy, Aditya Ramesh, Gabriel Goh, Sandhini Agarwal, Girish Sastry, Amanda Askell, Pamela Mishkin, Jack Clark, et al. Learning transferable visual models from natural language supervision. In *International conference on machine learning*, pp. 8748–8763. PMLR, 2021.

Kaushik Roy, Akhilesh Jaiswal, and Priyadarshini Panda. Towards spike-based machine intelligence with neuromorphic computing. *Nature*, 575(7784):607–617, 2019.

Ali Samadzadeh, Fatemeh Sadat Tabatabaei Far, Ali Javadi, Ahmad Nickabadi, and Morteza Haghir Chehreghani. Convolutional spiking neural networks for spatio-temporal feature extraction. *Neural Processing Letters*, 55(6):6979–6995, 2023.

Anh T Tran, Cuong V Nguyen, and Tal Hassner. Transferability and hardness of supervised classification tasks. In *Proceedings of the IEEE/CVF International Conference on Computer Vision*, pp. 1395–1405, 2019.

Xiangwen Wang, Xianghong Lin, and Xiaochao Dang. Supervised learning in spiking neural networks: A review of algorithms and evaluations. *Neural Networks*, 125:258–280, 2020.

Larry Wasserman. Bayesian model selection and model averaging. *Journal of mathematical psychology*, 44 (1):92–107, 2000.

Thomas Wolf, Lysandre Debut, Victor Sanh, Julien Chaumond, Clement Delangue, Anthony Moi, Pierric Cistac, Tim Rault, Rémi Louf, Morgan Funtowicz, Joe Davison, Sam Shleifer, Patrick von Platen, Clara Ma, Yacine Jernite, Julien Plu, Canwen Xu, Teven Le Scao, Sylvain Gugger, Mariama Drame, Quentin Lhoest, and Alexander M. Rush. Transformers: State-of-the-art natural language processing.

In *Proceedings of the 2020 Conference on Empirical Methods in Natural Language Processing: System Demonstrations*, pp. 38–45, Online, October 2020. Association for Computational Linguistics. URL https://www.aclweb.org/anthology/2020.emnlp-demos.6.

David Wolpert. On the use of evidence in neural networks. *Advances in neural information processing systems*, 5, 1992.

Yujie Wu, Lei Deng, Guoqi Li, Jun Zhu, Yuan Xie, and Luping Shi. Direct training for spiking neural networks: Faster, larger, better. In *Proceedings of the AAAI conference on artificial intelligence*, volume 33, pp. 1311–1318, 2019.

Man Yao, Guangshe Zhao, Hengyu Zhang, Yifan Hu, Lei Deng, Yonghong Tian, Bo Xu, and Guoqi Li. Attention spiking neural networks. *IEEE transactions on pattern analysis and machine intelligence*, 45 (8):9393–9410, 2023.

Jason Yosinski, Jeff Clune, Yoshua Bengio, and Hod Lipson. How transferable are features in deep neural networks? *Advances in neural information processing systems*, 27, 2014.

Kaichao You, Yong Liu, Jianmin Wang, and Mingsheng Long. Logme: Practical assessment of pre-trained models for transfer learning. In *International Conference on Machine Learning*, pp. 12133–12143. PMLR, 2021.

Kaichao You, Yong Liu, Ziyang Zhang, Jianmin Wang, Michael I Jordan, and Mingsheng Long. Ranking and tuning pre-trained models: a new paradigm for exploiting model hubs. *The Journal of Machine Learning Research*, 23(1):9400–9446, 2022.

Yuan Zhang, Jian Cao, Jue Chen, Wenyu Sun, and Yuan Wang. Razor snn: efficient spiking neural network with temporal embeddings. In *International Conference on Artificial Neural Networks*, pp. 411–422. Springer, 2023.

Lin Zhu, Xiao Wang, Yi Chang, Jianing Li, Tiejun Huang, and Yonghong Tian. Event-based video reconstruction via potential-assisted spiking neural network. In *Proceedings of the IEEE/CVF Conference on Computer Vision and Pattern Recognition*, pp. 3594–3604, 2022.

# A Appendix

## A.1 Details of pre-training SNNs

We use spiking version of MLP, spiking version of vanilla ResNet, SEW ResNet(Fang et al., 2021) and Attention spiking neural network(Yao et al., 2023) as candidate PTMs. On neuromorphic datasets, the accuracy on validate datasets is shown in Table 6. We use the same model architectures as in Fang et al. (2021). To adapt the neuromorphic dataset, we modified the input layer of the model, as shown in Table 7. The unchanged hyper-parameters are not notified in the table. The number of time steps on static datasets is 4 to be consistent with Fang et al. (2021) and 8 on neuromorphic datasets for better leverageing the temporal dimension of SNNs.

## A.2 Proofs

### A.2.1 Proof of Property 3

*Proof.* The logarithm of maximum evidence is calculated as equation 3,

$$\mathcal{L}_1^* = \frac{N}{2}\ln\beta^* + \frac{D}{2}\ln\alpha^* - \frac{N}{2}\ln 2\pi - \frac{\beta^*}{2}\|\mathbf{F}\boldsymbol{m} - \boldsymbol{t}\|^2 - \frac{\alpha^*}{2}\boldsymbol{m}^{\mathrm{T}}\boldsymbol{m} - \frac{1}{2}\ln\det\mathbf{A}, \tag{15}$$

where $\mathbf{A} = \alpha^*\mathbf{I} + \beta^*\mathbf{F}^{\mathrm{T}}\mathbf{F}, \boldsymbol{m} = \beta^*\mathbf{A}^{-1}\mathbf{F}^{\mathrm{T}}\boldsymbol{t}$.

Table 6: Pre-trained SNN models on ES-ImageNet dataset

| MODEL | TOP 1 ACC. | TOP 5 ACC. |
|-------|-----------|-----------|
| Spiking MLP-12 | 0.0180 | 0.0615 |
| Attention SNN | 0.1297 | 0.6710 |
| SEW ResNet-18 | 0.3443 | 0.5840 |
| SEW ResNet-34 | 0.3507 | 0.5922 |
| SEW ResNet-50 | 0.3140 | 0.5470 |
| SEW ResNet-101 | 0.3031 | 0.5347 |
| SEW ResNet-152 | 0.2615 | 0.4956 |
| Spiking ResNet-18 | 0.3156 | 0.5486 |
| Spiking ResNet-34 | 0.2376 | 0.4475 |
| Spiking ResNet-50 | 0.1661 | 0.3568 |

Table 7: Modification of the input layer of the models on neuromorphic datasets

| ORIGINAL (STATIC) | NEUROMORPHIC |
|-------------------|--------------|
| Conv2d(in_channels=3) | Conv2d(in_channels=2) |
| Batch normalization | Batch normalization |
| LIF | AvgPool2d(kernel_size=3) |
| MaxPool2d(kernel_size=3) | LIF |

When the input data $\mathbf{F}$ is multiplied by $q$ as $\widetilde{\mathbf{F}} = q\mathbf{F}$, let $\widetilde{\alpha}$ be $q^2\alpha^*$ and $\widetilde{\beta}$ be $\beta^*$, we have

$$\widetilde{\mathbf{A}} = q^2\alpha^*\mathbf{I} + \beta^* q\mathbf{F}^{\mathrm{T}} q\mathbf{F} = q^2\mathbf{A}, \tag{16}$$

$$\widetilde{\boldsymbol{m}} = \beta^*\widetilde{\mathbf{A}}^{-1}\widetilde{\mathbf{F}}^{\mathrm{T}}\boldsymbol{t} = \frac{1}{q}\boldsymbol{m}, \tag{17}$$

$$\det\widetilde{\mathbf{A}} = q^{2D}\det\mathbf{A}. \tag{18}$$

$\mathcal{L}_2^*$ is the maximum of logarithm of model evidence over $\alpha$ and $\beta$. Therefore we have

$$
\begin{aligned}
\mathcal{L}_2^* &\geq \mathcal{L}_2(\widetilde{\alpha}, \widetilde{\beta}) \\
&= \frac{N}{2}\ln\widetilde{\beta} + \frac{D}{2}\ln\widetilde{\alpha} - \frac{N}{2}\ln 2\pi - \frac{\widetilde{\beta}}{2}\left\|\widetilde{\mathbf{F}}\widetilde{\boldsymbol{m}} - \boldsymbol{t}\right\|^2 - \frac{\widetilde{\alpha}}{2}\widetilde{\boldsymbol{m}}^{\mathrm{T}}\widetilde{\boldsymbol{m}} - \frac{1}{2}\ln\det\widetilde{\mathbf{A}} \\
&= \frac{N}{2}\ln\beta^* + \frac{D}{2}\ln q^2\alpha^* - \frac{N}{2}\ln 2\pi - \frac{\beta^*}{2}\left\|(q\mathbf{F})\left(\frac{1}{q}\boldsymbol{m}\right) - \boldsymbol{t}\right\|^2 - \frac{q^2\alpha^*}{2}\left(\frac{1}{q}\boldsymbol{m}\right)^{\mathrm{T}}\left(\frac{1}{q}\boldsymbol{m}\right) - \frac{1}{2}\ln\det\left(q^2\mathbf{A}\right) \\
&= \frac{N}{2}\ln\beta^* + \frac{D}{2}\ln\alpha^* - \frac{N}{2}\ln 2\pi - \frac{\beta^*}{2}\|\mathbf{F}\boldsymbol{m} - \boldsymbol{t}\|^2 - \frac{\alpha^*}{2}\boldsymbol{m}^{\mathrm{T}}\boldsymbol{m} - \frac{1}{2}\ln\det\mathbf{A} \\
&= \mathcal{L}_1^*.
\end{aligned}
\tag{19}
$$

Similarly we have $\mathcal{L}_1^* \geq \mathcal{L}_2^*$. Therefore $\mathcal{L}_1^* = \mathcal{L}_2^*$. $\qquad\square$

### A.2.2 Proof of Theorem 1

*Proof.* Consider input data $\mathbf{F} \in \mathbb{R}^{N \times D}$ and targets $\boldsymbol{t} \in \mathbb{R}^N$. Let the approximated logarithm of maximum model evidence be $\mathcal{L}_1^A$. The singular value decomposition (SVD) of $\mathbf{F} = \mathbf{U}\boldsymbol{\Sigma}\mathbf{V}^{\mathrm{T}}$, where $\mathbf{U} \in \mathbb{R}^{N \times N}$ and $\mathbf{V} \in \mathbb{R}^{D \times D}$ are orthogonal matrix, $\sigma_1 \geq \cdots \geq \sigma_D \geq 0$ are singular values.

We first prove the case of property 1. When the input data $\mathbf{F}$ is stacked $q$ times as $\widetilde{\mathbf{F}} = [\mathbf{F}, \ldots, \mathbf{F}]$. From the proof of property 1 in You et al. (2022), the SVD of $\widetilde{\mathbf{F}}$ is $\widetilde{\mathbf{U}}\widetilde{\boldsymbol{\Sigma}}\widetilde{\mathbf{V}} = \mathbf{U}\left[\sqrt{q}\boldsymbol{\Sigma}, \mathbf{0}_{N \times (q-1)D}\right]\widetilde{\mathbf{V}}^{\mathrm{T}}$, where

$$\widetilde{\mathbf{V}} = \begin{bmatrix} \frac{1}{\sqrt{q}}\mathbf{V} & \cdots & \cdots \\ \vdots & \ddots & \vdots \\ \frac{1}{\sqrt{q}}\mathbf{V} & \cdots & \cdots \end{bmatrix} \in \mathbb{R}^{qD \times qD} \text{ is to product each element of } \mathbf{Q} = \begin{bmatrix} \frac{1}{\sqrt{q}} & \cdots & \cdots \\ \vdots & \ddots & \vdots \\ \frac{1}{\sqrt{q}} & \cdots & \cdots \end{bmatrix} \in \mathbb{R}^{q \times q} \text{ with } \mathbf{V}.$$

Here $\mathbf{Q}$ is an orthogonal matrix with first column $\left[\frac{1}{\sqrt{q}}, \frac{1}{\sqrt{q}}, \ldots, \frac{1}{\sqrt{q}}\right]^{\mathrm{T}}$, which can be obtained through Gram-Schmidt orthogonalization. The norm of stacked matrix $\|\widetilde{\mathbf{F}}\|_2 = \widetilde{\sigma}_1 = \sqrt{q}\sigma_1$ increases $\sqrt{q}$ times, where $\widetilde{\sigma}_1$ denotes the first singular value of $\widetilde{\mathbf{F}}$. $N$ is unchanged in this case. Therefore, $\widetilde{\lambda} = q\lambda$. By equation 12,

$$
\begin{aligned}
\widetilde{\boldsymbol{m}} &= \left(\widetilde{\lambda}\mathbf{I} + \widetilde{\mathbf{F}}^{\mathrm{T}}\widetilde{\mathbf{F}}\right)^{-1}\widetilde{\mathbf{F}}^{\mathrm{T}}\boldsymbol{t} = \widetilde{\mathbf{V}}\left(q\lambda\mathbf{I} + \widetilde{\boldsymbol{\Sigma}}^{\mathrm{T}}\widetilde{\boldsymbol{\Sigma}}\right)^{-1}\widetilde{\boldsymbol{\Sigma}}^{\mathrm{T}}\mathbf{U}^{\mathrm{T}}\boldsymbol{t} \\
&= \begin{bmatrix} \frac{1}{\sqrt{q}}\mathbf{V} & \cdots & \cdots \\ \vdots & \ddots & \vdots \\ \frac{1}{\sqrt{q}}\mathbf{V} & \cdots & \cdots \end{bmatrix} \begin{bmatrix} \frac{1}{q(\lambda+\sigma_1^2)} & & & & \\ & \ddots & & & \\ & & \frac{1}{q(\lambda+\sigma_D^2)} & & \\ & & & \frac{1}{q\lambda} & \\ & & & & \ddots \end{bmatrix} \begin{bmatrix} \sqrt{q}\boldsymbol{\Sigma}^{\mathrm{T}} \\ \mathbf{0}_{(q-1)D \times N} \end{bmatrix}\mathbf{U}^{\mathrm{T}}\boldsymbol{t} \\
&= \begin{bmatrix} \frac{1}{q}\mathbf{V}\left(\lambda\mathbf{I} + \boldsymbol{\Sigma}^{\mathrm{T}}\boldsymbol{\Sigma}\right)^{-1}\boldsymbol{\Sigma}^{\mathrm{T}}\mathbf{U}^{\mathrm{T}}\boldsymbol{t} \\ \vdots \\ \frac{1}{q}\mathbf{V}\left(\lambda\mathbf{I} + \boldsymbol{\Sigma}^{\mathrm{T}}\boldsymbol{\Sigma}\right)^{-1}\boldsymbol{\Sigma}^{\mathrm{T}}\mathbf{U}^{\mathrm{T}}\boldsymbol{t} \end{bmatrix} = \frac{1}{q}\begin{bmatrix} \boldsymbol{m} \\ \vdots \\ \boldsymbol{m} \end{bmatrix}.
\end{aligned}
\tag{20}
$$

Therefore we have $\widetilde{\beta}_0 = \beta_0, \widetilde{\alpha}_0 = \widetilde{\lambda}\widetilde{\beta}_0 = q\alpha_0$. From equation 3, the approximated logarithm of $\widetilde{\mathbf{F}}$ and $\boldsymbol{t}$ is

$$
\begin{aligned}
\mathcal{L}_2^A &= \frac{N}{2}\ln\widetilde{\beta}_0 + \frac{qD}{2}\ln\widetilde{\alpha}_0 - \frac{N}{2}\ln 2\pi - \frac{\widetilde{\beta}_0}{2}\left\|\widetilde{\mathbf{F}}\widetilde{\boldsymbol{m}} - \boldsymbol{t}\right\|^2 - \frac{\widetilde{\alpha}_0}{2}\widetilde{\boldsymbol{m}}^{\mathrm{T}}\widetilde{\boldsymbol{m}} - \frac{1}{2}\ln\det\widetilde{\mathbf{A}} \\
&= \frac{N}{2}\ln\beta_0 + \frac{qD}{2}\ln q\alpha_0 - \frac{N}{2}\ln 2\pi - \frac{\beta_0}{2}\|\mathbf{F}\boldsymbol{m} - \boldsymbol{t}\|^2 - \frac{\alpha_0}{2}\boldsymbol{m}^{\mathrm{T}}\boldsymbol{m} \\
&\quad - \frac{1}{2}\ln\det\begin{bmatrix} q(\alpha_0 + \beta_0\sigma_1^2) & & & & \\ & \ddots & & & \\ & & q(\alpha_0 + \beta_0\sigma_D^2) & & \\ & & & q\alpha_0 & \\ & & & & \ddots \end{bmatrix} \\
&= \frac{N}{2}\ln\beta_0 + \frac{D}{2}\ln\alpha_0 + \frac{(q-1)D}{2}\ln\alpha_0 + \frac{qD}{2}\ln q - \frac{N}{2}\ln 2\pi - \frac{\beta_0}{2}\|\mathbf{F}\boldsymbol{m} - \boldsymbol{t}\|^2 - \frac{\alpha_0}{2}\boldsymbol{m}^{\mathrm{T}}\boldsymbol{m} - \frac{1}{2}\ln\det\mathbf{A} \\
&\quad - \frac{1}{2}\ln q^{qD}\alpha_0^{(q-1)D} \\
&= \mathcal{L}_1^A.
\end{aligned}
\tag{21}
$$

Then we prove the case of property 2. When the input data $\mathbf{F}$ is padded with zeros as $\widetilde{\mathbf{F}} = [\mathbf{F}, \mathbf{0}]$. Its SVD is $\mathbf{U}\left[\boldsymbol{\Sigma}, \mathbf{0}_{N \times (q-1)D}\right]\begin{bmatrix} \mathbf{V} & \\ & \mathbf{V}_1 \end{bmatrix}^{\mathrm{T}}$, where $\mathbf{V}_1$ is an orthogonal matrix. The norm $\|\widetilde{\mathbf{F}}\|_2 = \widetilde{\sigma}_1$ keeps the same and $N$ is unchanged as well. Therefore, $\widetilde{\lambda}$ is unchanged for $\widetilde{\mathbf{F}}$.

From the proof of property 1 in You et al. (2022), we have

$$
\widetilde{m} = \begin{bmatrix} \mathbf{V} & \\ & \mathbf{V}_1 \end{bmatrix} \begin{bmatrix} \frac{1}{\lambda+\sigma_1^2} & & & \\ & \ddots & & \\ & & \frac{1}{\lambda+\sigma_D^2} & \\ & & & \frac{1}{\lambda} \\ & & & & \ddots \end{bmatrix} \begin{bmatrix} \boldsymbol{\Sigma}^{\mathrm{T}} \\ \mathbf{0}_{(q-1)D \times N} \end{bmatrix} \mathbf{U}^{\mathrm{T}} \boldsymbol{t}
\tag{22}
$$

$$
= \begin{bmatrix} \mathbf{V} \left( \lambda \mathbf{I} + \boldsymbol{\Sigma}^{\mathrm{T}} \boldsymbol{\Sigma} \right)^{-1} \boldsymbol{\Sigma}^{\mathrm{T}} \mathbf{U}^{\mathrm{T}} \boldsymbol{t} \\ \mathbf{0}_{(q-1)D \times 1} \end{bmatrix} = \begin{bmatrix} m \\ \mathbf{0}_{(q-1)D \times 1} \end{bmatrix}.
$$

Therefore we have $\widetilde{\beta}_0 = \beta_0, \widetilde{\alpha}_0 = \widetilde{\lambda}\widetilde{\beta}_0 = \alpha_0$. From equation 3,

$$
\mathcal{L}_2^A = \frac{N}{2} \ln \widetilde{\beta}_0 + \frac{qD}{2} \ln \widetilde{\alpha}_0 - \frac{N}{2} \ln 2\pi - \frac{\widetilde{\beta}_0}{2} \left\| \widetilde{\mathbf{F}}\widetilde{m} - \boldsymbol{t} \right\|^2 - \frac{\widetilde{\alpha}_0}{2} \widetilde{m}^{\mathrm{T}}\widetilde{m} - \frac{1}{2} \ln \det \widetilde{\mathbf{A}}
$$

$$
= \frac{N}{2} \ln \beta_0 + \frac{qD}{2} \ln \alpha_0 - \frac{N}{2} \ln 2\pi - \frac{\beta_0}{2} \|\mathbf{F}m - \boldsymbol{t}\|^2 - \frac{\alpha_0}{2} m^{\mathrm{T}}m - \frac{1}{2} \ln \det \begin{bmatrix} (\alpha_0 + \beta_0\sigma_1^2) & & & & \\ & \ddots & & & \\ & & (\alpha_0 + \beta_0\sigma_D^2) & & \\ & & & \alpha_0 & \\ & & & & \ddots \end{bmatrix}
$$

$$
= \frac{N}{2} \ln \beta_0 + \frac{D}{2} \ln \alpha_0 + \frac{(q-1)D}{2} \ln \alpha_0 - \frac{N}{2} \ln 2\pi - \frac{\beta_0}{2} \|\mathbf{F}m - \boldsymbol{t}\|^2 - \frac{\alpha_0}{2} m^{\mathrm{T}}m - \frac{1}{2} \ln \det \mathbf{A} - \frac{1}{2} \ln \alpha_0^{(q-1)D}
$$

$$
= \mathcal{L}_1^A.
\tag{23}
$$

We prove the case if property 3. When the input data $\mathbf{F}$ is multiplied by scalar $q > 0$ as $\widetilde{\mathbf{F}} = q\mathbf{F}$. $\|\widetilde{\mathbf{F}}\|_2 = \widetilde{\sigma}_1 = q\sigma_1$ increases $q$ times and $N$ is unchanged. Therefore, $\widetilde{\lambda}$ increases $q^2$ times for $\widetilde{\mathbf{F}}$.

From the proof of property 3, $\widetilde{\mathbf{A}} = q^2\mathbf{A}, \widetilde{m} = \frac{1}{q}m$. Therefore we have $\widetilde{\beta}_0 = \beta_0, \widetilde{\alpha} = \widetilde{\lambda}\widetilde{\beta} = q^2\alpha$. From equation 3,

$$
\mathcal{L}_2^A = \frac{N}{2} \ln \widetilde{\beta}_0 + \frac{D}{2} \ln \widetilde{\alpha}_0 - \frac{N}{2} \ln 2\pi - \frac{\widetilde{\beta}_0}{2} \left\| \widetilde{\mathbf{F}}\widetilde{m} - \boldsymbol{t} \right\|^2 - \frac{\widetilde{\alpha}_0}{2} \widetilde{m}^{\mathrm{T}}\widetilde{m} - \frac{1}{2} \ln \det \widetilde{\mathbf{A}}
$$

$$
= \frac{N}{2} \ln \beta_0 + \frac{D}{2} \ln q^2\alpha_0 - \frac{N}{2} \ln 2\pi - \frac{\beta_0}{2} \left\| (q\mathbf{F}) \left(\frac{1}{q}m\right) - \boldsymbol{t} \right\|^2 - \frac{q^2\alpha_0}{2} \left(\frac{1}{q}m\right)^{\mathrm{T}} \left(\frac{1}{q}m\right) - \frac{1}{2} \ln \det \left(q^2\mathbf{A}\right)
$$

$$
= \frac{N}{2} \ln \beta_0 + \frac{D}{2} \ln \alpha_0 - \frac{N}{2} \ln 2\pi - \frac{\beta_0}{2} \|\mathbf{F}m - \boldsymbol{t}\|^2 - \frac{\alpha_0}{2} m^{\mathrm{T}}m - \frac{1}{2} \ln \det \mathbf{A}
$$

$$
= \mathcal{L}_1^A.
\tag{24}
$$

$\square$

### A.2.3 Proof of case $t = \hat{t} = 0$

For $t = \hat{t} = 0$, we have $o(t) = 0, 1 \le t \le T$ and thus $u(t) = \mathrm{e}^{-\frac{\Delta t}{\tau}} u(t-1) + \boldsymbol{\theta}^{\mathrm{T}} \boldsymbol{f}(t)$. Note that $u(0) = 0$, it can be conducted that

$$
u(1) = \boldsymbol{\theta}^{\mathrm{T}} \boldsymbol{f}(1) \le u_{th}
$$

$$
u(T_0) = \boldsymbol{\theta}^{\mathrm{T}} \left( \sum_{t=1}^{T_0} \mathrm{e}^{-(T_0-t)\frac{\Delta t}{\tau}} \boldsymbol{f}(t) \right) \le u_{th}, 1 \le T_0 \le T
\tag{25}
$$

By summing $u_i(T_0)$ from 1 to $T - 1$, we have

$$
\sum_{T_0=1}^{T-1} u(T_0) = \boldsymbol{\theta}^{\mathrm{T}} \left( \sum_{T_0=1}^{T-1} \left( \boldsymbol{f}(T_0) \sum_{t=0}^{T-T_0-1} \mathrm{e}^{-t\frac{\Delta t}{\tau}} \right) \right) \le (T-1)u_{th}.
\tag{26}
$$

This is equivalent to

$$\boldsymbol{\theta}^{\mathrm{T}} \left( \sum_{T_0=1}^{T-1} \left( \boldsymbol{f}(T_0) \left( 1 - \mathrm{e}^{-(T-T_0)\frac{\Delta t}{\tau}} \right) \right) \right) \leq \left( 1 - \mathrm{e}^{-\frac{\Delta t}{\tau}} \right) (T-1) u_{th}. \tag{27}$$

Note that we have

$$u(T) = \boldsymbol{\theta}^{\mathrm{T}} \left( \sum_{T_0=1}^{T} \mathrm{e}^{-(T-T_0)\frac{\Delta t}{\tau}} \boldsymbol{f}(T_0) \right) \leq u_{th}. \tag{28}$$

By adding Equation 27 and 28 together, we have $\boldsymbol{\theta}^{\mathrm{T}} \left( \sum_{T_0=1}^{T} \boldsymbol{f}(T_0) \right) \leq \left( T - (T-1)\mathrm{e}^{-\frac{\Delta t}{\tau}} \right) u_{th}$, which is

$$\boldsymbol{\theta}^{\mathrm{T}} \bar{\boldsymbol{f}} = \frac{1}{T} \sum_{t=1}^{T} \boldsymbol{f}(t) \leq \left( 1 - \frac{T-1}{T} \mathrm{e}^{-\frac{\Delta t}{\tau}} \right) u_{th} \leq u_{th}. \tag{29}$$

### A.3  Validation of approximated maximum evidence

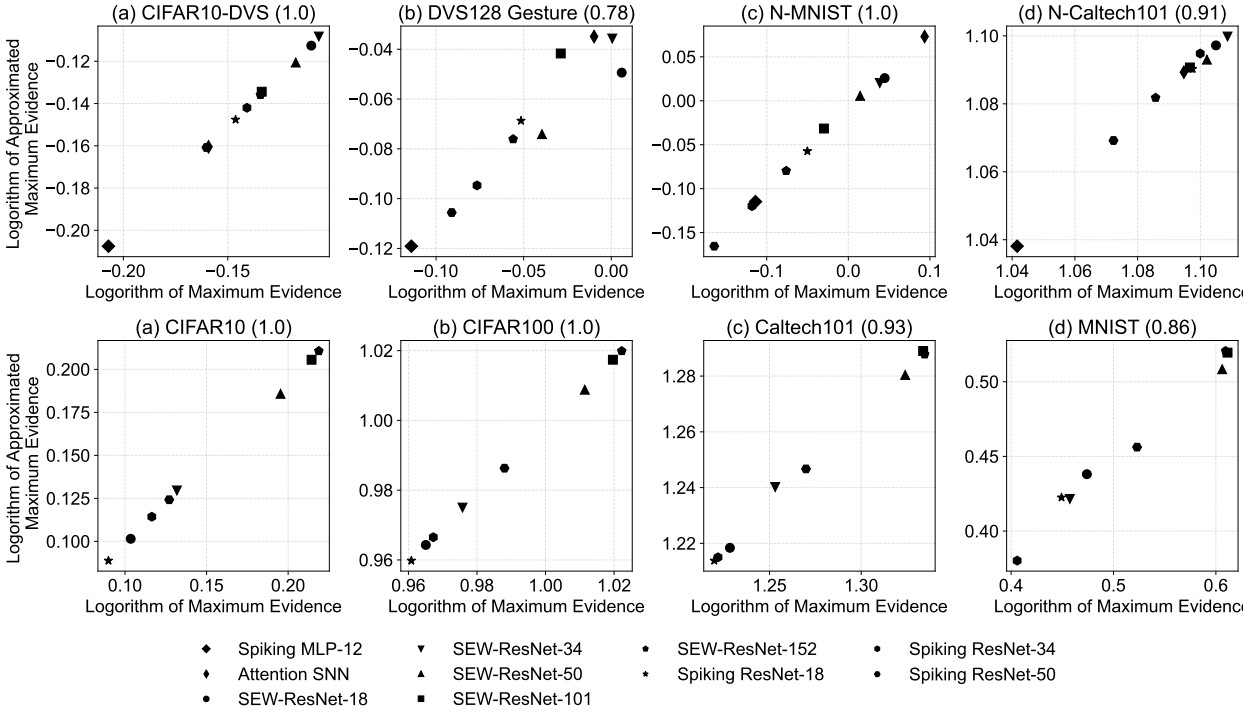

Figure 6:  Results of scores given by **approximated method (y-axis) vs. MacKay's method (x-axis)**. Kendall correlation coefficients are labeled in the title of each subplot after the dataset name.

We demonstrated a strong correlation between the approximated maximum evidence and those obtained using MacKay's algorithm (Figure 6). It is worth noting that the correlation is relatively low on the DVS128Gesture dataset. We believe this is because the DVS128 Gesture is a very small dataset, with only 1176 samples in the training set (Amir et al., 2017), which contradicts our assumption of the number of samples being much greater than the dimension of features (as shown in Figure 3).

### A.4  Study on error range

Here we discuss the error range of test set accuracy in the experiments. Due to computational resource limitations, we only conducted experiments on a single typical dataset, DVS128 Gesture. Each pre-trained

model was fine-tuned on this dataset with 8 different random seeds and grid searched with learning rates ranging from 1$e$-1 to 1$e$-3 and weight decay ranging from 1$e$-5 to 1$e$-8. Each model is fine-tuned 200 epochs. Based on the results, we calculated the mean and standard deviation (std) of the test set accuracy on this dataset, which are presented in Table 8. Additionally, to better illustrate the impact of the error range in test set accuracy on transferability assessment, we also present the corresponding results in Figure 7.

Table 8: Error range of test set accuracy on DVS128 Gesture dataset

| MODEL | NCE | LEEP | MEAF | TEST ACC. (STD) |
|---|---|---|---|---|
| Spiking MLP-12 | -2.10835 | -2.25607 | -0.11412 | 0.306 (0.009) |
| Attention SNN | -2.06354 | -2.36722 | -0.00981 | 0.615 (0.022) |
| SEW ResNet-18 | -2.16645 | -2.18893 | 0.00596 | 0.656 (0.012) |
| SEW ResNet-34 | -2.00966 | -2.07468 | 0.00055 | 0.635 (0.008) |
| SEW ResNet-50 | -2.26797 | -2.26722 | -0.03956 | 0.622 (0.014) |
| SEW ResNet-101 | -2.17705 | -2.18028 | -0.02884 | 0.639 (0.017) |
| SEW ResNet-152 | -2.14735 | -2.16500 | -0.05603 | 0.565 (0.013) |
| Spiking ResNet-18 | -2.28786 | -2.27276 | -0.05153 | 0.523 (0.012) |
| Spiking ResNet-34 | -2.23009 | -2.26244 | -0.07663 | 0.480 (0.015) |
| Spiking ResNet-50 | -2.18648 | -2.29238 | -0.09107 | 0.497 (0.012) |

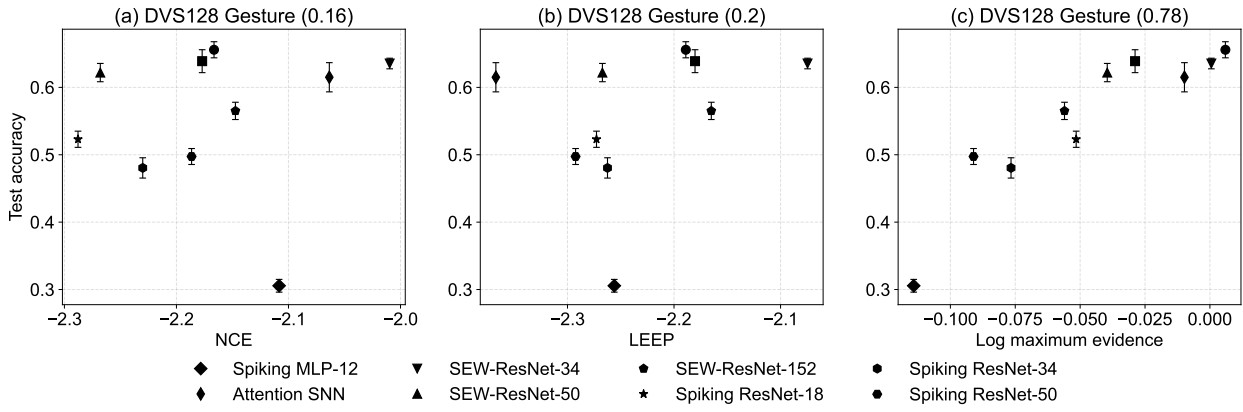

Figure 7: Results of **test set accuracy vs. assessment score (x-axis)** with error bars. Kendall correlation coefficients are labeled in the title of each subplot after the dataset name.

## A.5 Ablation study

Here we investigate the effectiveness of averaging the features. We selected three alternative ways for constructing feature vectors of shape $N \times D$ from SNN feature maps of shape $N \times T \times D$, where $N$ is the dataset size, $T$ is the number of time steps and $N$ is the dimension of feature of each time step. We use these feature vectors to assess transferability with MacKay's method and compare the results with those from MEAF. The three methods we selected are:

1. **LAST**: Selecting the feature vector from the last time step,

2. **MAX**: Applying max-pooling along the time dimension to the feature vectors,

3. **PCA**: Using PCA to reduce the SNN feature map from shape $N \times T \times D$ to $N \times D$,

and the results are presented in Table 9.

The MEAF method performs optimally on most datasets, achieving the highest average Kendall correlation coefficient. On the CIFAR10-DVS and N-Caltech101 datasets, it also achieves the second-best results, with a gap of less than 0.04 from the optimal results. This demonstrates the effectiveness of average feature construction. Other methods to aggregate feature from multi-time steps, such as using a convolution layer along the time dimension may have better performance ; however, these involve additional learnable parameters that need to be trained or set, making them more complex compared to the current approaches. We leave this as a potential direction for future research.

Table 9: Kendall coefficient of transferability calculated by alternative ways to construct feature vector and MEAF

| DATASET | LAST | MAX | PCA | MEAF |
|---|---|---|---|---|
| DVS128 | 0.69 | 0.73 | 0.69 | **0.78** |
| C10-DVS | 0.91 | 0.91 | **1.00** | 0.96 |
| N-C101 | **0.91** | 0.87 | 0.69 | 0.87 |
| N-MNIST | **0.87** | 0.78 | 0.73 | **0.87** |
| C10 | 0.79 | 0.79 | **0.86** | **0.86** |
| C100 | **0.71** | **0.71** | **0.71** | **0.71** |
| MNIST | 0.71 | 0.71 | 0.71 | **0.79** |
| C101 | 0.43 | 0.43 | **0.50** | **0.50** |
| **AVERAGE** | 0.75 | 0.74 | 0.74 | **0.79** |

