# OpenReview forum: "Bayesian Transferability Assessment for Spiking Neural Networks"
_TMLR — Accepted by TMLR_

### Review · Reviewer_HHBr · 2024-12-20

**Summary Of Contributions:**

The paper proposes an approach for model selection for spiking neural networks (SNNs). The approach is based on a commonly used technique in ANNs where features from pretrained networks are passed to a fully connected layer for prediction (linear probing). The model evidence can then be calculated analytically using Mackay’s algorithm The paper proposes the use of time-averaged feature maps from the pretrained SNNs to approximate the linear probing process of ANNs. It also proposes a method to accelerate the convergence of Mackay’s algorithm using an approximation. Through multiple experiments, the paper shows that the proposed approach can compute the transferability of the pre-trained SNN effectively.

**Audience:**

No

**Claims And Evidence:**

No

**Requested Changes:**

See weaknesses

**Strengths And Weaknesses:**

**Strengths:**
1. The paper is largely well written and was easy to follow, although there is scope for more clarity.
2. The problem of transferability assessment for SNNs is indeed untackled and may require some thinking from the community, especially as pretraining large foundational models seems to be becoming a norm.
3. The experimental results, despite their limitations, provide initial validation of the proposed method and demonstrate its potential effectiveness.

**Weaknesses:**
1. **Claims:**
    - The primary motivation that the paper states is that SNN models are much harder to optimize than ANN due to non-differentiability of the spike. However, the primary motivation for pretraining + finetuning in ANNs is to alleviate the issue of training data scarcity. In the context of SNNs, the non-differentiability issue will still remain to some degree during finetuning even after picking the right pretrained model. The paper should state explicitly how optimization issues are resolved starting from pretrained model, as opposed to starting from scratch.
    - The limited availability of large pretrained models for SNNs reduces the practical impact and interest of the proposed solution. Given the fact that commonly used pretrained backbones do not differ significantly in test performance, it is difficult to think of a scenario where one would run the proposed algorithm before picking which backbone to use.

2. **Clarity:** While the paper is mostly clear, it could benefit from having an Algorithm that describes the process.

3. **Method Critique:** My main critique of the method is that the authors propose one approach for approximating linear probing- use time-averaged feature maps from the SNN. They provide a proof of why this is a linear transform. However, there are many other potential solutions that have not been discussed:
    - Using features from the last time step
    - Applying a linear dim-reduction technique such as PCA on the feature maps from all the layers
    - Adding a temporal convolution layer to get weighted time-average of feature maps
    - Pooling operation on time

4. **Approximated Method Clarity:** The approximated method (for model evidence) is not clear to me. Why do equations 11-14 accelerate the convergence? The last paragraph of Section 3 says: “Moreover, with more reasonable prior values, MacKay’s method can be effectively accelerated”. What is “more reasonable” in this sense? How is this acceleration different from approximated method?

5. **Choice of Baselines:** The choice of baselines isn’t motivated. Why are only information-theoretic methods used for comparison? Have these been used previously for SNN applications? Also, the baselines seem to be doing pretty bad in most of the experiments. Are these the right methods to calculate transferability?

6. **Experiments and Results:** Several points require clarification:
    - Pretrained models are all ResNets. The paper would benefit from adding attention-based architectures such as transformers [1].
    - From Figure 2, it looks like if the test accuracy of finetuning pretrained model is high, then transferability measure loses its effectiveness. This can be a big issue with large pretrained models that can have high test performance on target datasets.
    - While the paper shows # of iterations for Mackay in appendix, there is no comparison with # of iterations with approximated method. How much time is being saved because of using the approximated method? Is this a function of dataset?
    - Looks like the approximated method has highly varied performance w.r.t. computing transferability. For some datasets, it is very close to Mackay but for some others- DVS Gesture and MNIST, there is a big drop. The paper states that it could be due to DVS Gesture being very small w.r.t. the number of features (this line should be moved to the main section). Then how realistic is the assumption of N>>D? Finetuning is best used in limited training data regime, then why would the approximated method be useful there? Also, why does MNIST show a drop?

---

> ### Author Response · Authors · 2025-01-20
> **Response to the reviewer (1-2) (claims)**
>
> Dear reviewer,
>
> Thank you for your careful and kind review. Your comments are meaningful and helpful for our work. Here are our responses to your concerns.
>
> 1. The primary motivation that the paper states is that SNN models are much harder to optimize than ANN due to non-differentiability of the spike. However, the primary motivation for pretraining + finetuning in ANNs is to alleviate the issue of training data scarcity. In the context of SNNs, the non-differentiability issue will still remain to some degree during finetuning even after picking the right pretrained model. The paper should state explicitly how optimization issues are resolved starting from pretrained model, as opposed to starting from scratch.
>
>
> ### Response
> Thank you for your question. The “pre-training + fine-tuning” paradigm in ANNs is indeed primarily used to address the issue of scarce training data. However, many studies have also shown that models pre-trained on large-scale datasets tend to perform better[1] or converge more efficiently[2] on downstream tasks. Some research related to SNNs has also demonstrated that supervised/unsupervised pre-trained SNN models have higher optimization efficiency when fine-tuned on downstream tasks[3, 4]. Therefore, the “pre-training + fine-tuning” paradigm can help alleviate the optimization challenges of SNNs to some extent.
>
>
> On the other hand, SNNs require multiple time steps of forward pass to generate results, which means that SNN models with the same number of parameters will take several times longer to train compared to ANNs. As a result, training large-scale SNNs consumes more computational resources and time than training ANNs. Effectively utilizing pre-trained models and performing only a small amount of fine-tuning can significantly reduce the resources and time needed for training SNNs on the target task, thereby facilitating the broader application of SNNs.
>
> To better clarify on this problem, we have add more explanation in the end of the second paragraph of the Intro. You can check it in the revised manuscript. Hope that this could alleviate you concern to some extend and we are looking forward to further discussion.
>
>
> [1] He, Kaiming, Ross Girshick, and Piotr Dollár. "Rethinking imagenet pre-training." Proceedings of the IEEE/CVF international conference on computer vision. 2019.
>
>
> [2] Kornblith, Simon, Jonathon Shlens, and Quoc V. Le. "Do better imagenet models transfer better?." Proceedings of the IEEE/CVF conference on computer vision and pattern recognition. 2019.
>
>
> [3] Lin, Yihan, et al. "Rethinking pretraining as a bridge from anns to snns." IEEE Transactions on Neural Networks and Learning Systems (2022).
>
>
> [4] Lee, Chankyu, et al. "Training deep spiking convolutional neural networks with STDP-based unsupervised pre-training followed by supervised fine-tuning." Frontiers in neuroscience 12 (2018): 435.
>
>
> 2. The limited availability of large pretrained models for SNNs reduces the practical impact and interest of the proposed solution. Given the fact that commonly used pretrained backbones do not differ significantly in test performance, it is difficult to think of a scenario where one would run the proposed algorithm before picking which backbone to use.
>
> ### Response
> This is indeed a problem as available large-scale pretrained SNN models are relatively limited compared with ANNs.
> However, with the development of the SNN field, we are also seeing an increasing number of studies on pretrained SNN models, such as SNN-based LLM by auto-regressive pretraining [1], masked langauge modeling[2] and vision encoders by contrastive pretraining[3]. We believe that with the continued growth of the SNN field, there will inevitably be a large number of pretrained SNN models in the future. Therefore, from a forward-looking perspective, it is meaningful to explore this issue.
>
> [1]Zhu, Rui-Jie, et al. "Spikegpt: Generative pre-trained language model with spiking neural networks." arXiv preprint arXiv:2302.13939 (2023).
>
>
> [2]Su Q, Mei S, Xing X, et al. SNN-BERT: Training-efficient Spiking Neural Networks for energy-efficient BERT[J]. Neural Networks, 2024, 180: 106630.
>
>
> [3]Li T, Liu W, Lv C, et al. Spikeclip: A contrastive language-image pretrained spiking neural network[J]. arXiv preprint arXiv:2310.06488, 2023.

---

> ### Author Response · Authors · 2025-01-20
> **Response to the reviewer (3-4) (Clarity & Method Critique)**
>
> 3. While the paper is mostly clear, it could benefit from having an Algorithm that describes the process.
>
> ### Response
> Thank you for your advice. We have added an Algorithm section at the end of section 3 on page 7. You can check it in the revised manuscript.
>
> 4. Method Critique: My main critique of the method is that the authors propose one approach for approximating linear probing- use time-averaged feature maps from the SNN. They provide a proof of why this is a linear transform. However, there are many other potential solutions that have not been discussed:
> - Using features from the last time step
> - Applying a linear dim-reduction technique such as PCA on the feature maps from all the layers
> - Adding a temporal convolution layer to get weighted time-average of feature maps
> - Pooling operation on time
>
>
> ### Response
> This is indeed a very good question as other kinds of construction of feature vector can also be used instead of the averaged feature.
> We have added a discussion on this issue in section A.5 of the appendix in the revised manuscript.
> We conducted experiments with three methods:
> - LAST: using the last time step
> - MAX: using max-pooling over time
> - PCA: using PCA for dimensionality reduction
>
> |DATASET | LAST | MAX | PCA | MEAF |
> | ---- | ---- | ----| ----| ----|
> | DVS128 Gesture | 0.69 | 0.73 | 0.69 | **0.78** |
> |CIFAR10-DVS | 0.91 | 0.91 | **1.00** | 0.96 |
> |N-Caltech101 | **0.91** | 0.87 | 0.69 | 0.87 |
> |N-MNIST | **0.87** | 0.78 | 0.73 | **0.87** |
> |CIFAR10 | 0.57 | 0.57 | **0.64** | **0.64** |
> |CIFAR100 | **0.64** | **0.64** | **0.64** | **0.64** |
> |MNIST | 0.64 | 0.64 | 0.64 | **0.71** |
> |Caltech101 | 0.69 | 0.69 | **0.76** | **0.76** |
> | Average| 0.74 | 0.73 | 0.72 | **0.79**|
>
> For each method, we computed transferability on the feature vectors generated using MacKay's method.
> We compare the results with the MEAF method that uses time-averaged features.
> The results are shown in the table above.
> The experimental results show that, on most datasets (6/8), the results from MEAF are the best, and on the remaining two datasets, the difference from the optimal result is within 0.04.
> This indicates that the selection of average features in MEAF is of effectiveness.
> The results are also shown in Table 6 in the revised manuscript.
>
> We give a further discussion on these alternative methods here.
>
>
> (1) For the method using the last time step, it loses most of the information, retaining only the information from the final time step, resulting in poor performance.
>
> (2) For max-pooling, since the features of an SNN are composed of 01 spikes, the max-pooling process may discard too much information.
>
> (3) The PCA method does effectively compress the information, but it introduces additional computational overhead. This is especially true for large-scale datasets, where the speed of PCA dimensionality reduction is relatively slow, which may impact the efficiency of our method.
>
> (4) The method of applying time convolution to weight the features at each time step could potentially yield better results, as our time-averaging method is essentially a special case of this method with equal weights. However, this method introduces additional parameters that need to be trained, thus increasing the complexity of the problem. We believe this could be a direction for future research to explore further.

---

> ### Author Response · Authors · 2025-01-20
> **Response to the reviewer (5-6) (Approximated Method Clarity & Choice of Baselines)**
>
> 5. The approximated method (for model evidence) is not clear to me. Why do equations 11-14 accelerate the convergence? The last paragraph of Section 3 says: “Moreover, with more reasonable prior values, MacKay’s method can be effectively accelerated”. What is “more reasonable” in this sense? How is this acceleration different from approximated method?
>
>
> ### Response
> Thank you for pointing out the issue, and sorry for the ambiguity in the original wording. The approximated method can serve as an approximation of MacKay's method and does not require iterations to produce results, thus it can reduce the time needed for the MacKay's method. However, it has not been verified whether using the approximated method's results as the initial value of MacKay's method could accelerate the convergence of MacKay's method. Therefore, to make the expression more precise, we have revised the original text and removed this part.
>
> 6. The choice of baselines isn’t motivated. Why are only information-theoretic methods used for comparison? Have these been used previously for SNN applications? Also, the baselines seem to be doing pretty bad in most of the experiments. Are these the right methods to calculate transferability?
>
>
> ### Response 6
> Thank you for your question.
>
> NCE and LEEP are transferability assessment methods based on information theory. From a theoretical perspective, these methods do not impose any specific requirements on the neural network architecture (ANN or SNN); they only need the prediction results provided by the model (probability distribution or pseudo-labels) to assess transferability. Therefore, although these methods were initially proposed on ANNs, they are also applicable to SNNs.
>
> We choose NCE and LEEP as baseline because of basically two reasons. The first is that they are the commonly used baselines of transferability assessment[1, 2]. Second, they can be directly applied to SNNs without other modifications.
>
> [1] You, Kaichao, et al. "Logme: Practical assessment of pre-trained models for transfer learning." International Conference on Machine Learning. PMLR, 2021.
> [2] You, Kaichao, et al. "Ranking and tuning pre-trained models: A new paradigm for exploiting model hubs." Journal of Machine Learning Research 23.209 (2022): 1-47.

---

> ### Author Response · Authors · 2025-01-20
> **Response to the reviewer (7-8) (Experiments and Results)**
>
> 7. Pretrained models are all ResNets. The paper would benefit from adding attention-based architectures such as transformers
>
> ### Response
> Thank you for your suggestion. We have added more models to the experiments on the neuromorphic datasets, including an SNN model based on the attention mechanism [1] and a simple SNN-based MLP.  After adding the new models, the changes of Kendall coefficients are:
> - CIFAR10-DVS (0.93 $\xrightarrow{}$ 0.96)
> - DVS128 Gesture (0.84 $\xrightarrow{}$ 0.78)
> - N-Caltech101 (0.93 $\xrightarrow{}$ 0.87)
> - N-MNIST (0.86 $\xrightarrow{}$ 0.87)
>
> The new experimental results can also be found in Fig. 1 and 4. You can refer to them in the revised manuscript.
>
> [1] Yao, Man, et al. "Attention spiking neural networks." IEEE transactions on pattern analysis and machine intelligence 45.8 (2023): 9393-9410.
>
>
> 8. From Figure 2, it looks like if the test accuracy of finetuning pretrained model is high, then transferability measure loses its effectiveness. This can be a big issue with large pretrained models that can have high test performance on target datasets
>
> ### Response
> Thank you for your question. When the model accuracy is relatively high, the assessment of transferability does have some inaccuracies, but based on the experimental results, we still find these results meaningful.
>
>
> First, for the four datasets in Fig 2, both MNIST and Caltech101 accurately identified the models with the highest accuracy, and in CIFAR10 and CIFAR100, the top 3 scored models also included the highest-accuracy model.
> The transferability assessment method is designed to reduce the time required for selecting the best model for downstream tasks, which requires finetuning and grid searching on all the PTMs without the assessment method.
> In practical applications, one can select the top K models based on the assessed scores for subsequent experiments without the need to finetune and select from all models.
> In this sense, the best models are all included in the top K, so our method can still be effective.
>
>
> Secondly, in Fig 2, the performance differences between the models are not large, and the range of the x-axis is much smaller than in Fig 1, so the differences in transferability between the models are actually not significant.
> In this case, it makes sense that the correlation coefficient is smaller compared to Fig 1.
>
>
> Third, MNIST is the dataset with the highest accuracy.
> From the results on MNIST, we can see that the model did not fail when the accuracy was high but accurately identified the top 3 models with the best transferability.
> This indicates that the method did not experience a significant decline in performance when accuracy was high.

---

> ### Author Response · Authors · 2025-01-20
> **Response to the reviewer (9-10) (Experiments and Results)**
>
> 9. While the paper shows \# of iterations for Mackay in appendix, there is no comparison with \# of iterations with approximated method. How much time is being saved because of using the approximated method? Is this a function of dataset?
>
> ### Response
> Thank you for your question. We shows \# of iterations of MacKay's method in the appendix (of the previous manuscript) to show that our approximated method could save time. In fact, because that our approximated method do not need any iteration, the \# of iteration of the approximated method is exactly 0. Therefore, we didn't show it in the manuscript.
>
>
> To make it more clear, we move this part to the main body of the paper and add more descriptions about this. You can check it in the revised manuscript.
>
>
> 10. Looks like the approximated method has highly varied performance w.r.t. computing transferability. For some datasets, it is very close to Mackay but for some others- DVS Gesture and MNIST, there is a big drop. The paper states that it could be due to DVS Gesture being very small w.r.t. the number of features (this line should be moved to the main section). Then how realistic is the assumption of N$>>$D? Finetuning is best used in limited training data regime, then why would the approximated method be useful there? Also, why does MNIST show a drop?
>
>
> ### Response
> Thank you for your comment.
> For the first question.
> In fact, DVS128 Gesture dataset is very small. It only have 1176 samples in the train set and 288 samples in the test set. The dimension of feature (D) of SEW-ResNet-101 and SEW-ResNet-152 is 2048, which is larger than 1176.
> Therefore, on this dataset, the assumption of $N >> D$ is not satisfied.
> From the result of Fig. 3, the approximated method differs a a lot with MacKay's method.
>
> The purpose of the proposed approximation method is to reduce the computation time of MacKay’s method. When$N$ is small, the time complexity of each iteration of the MacKay method is $O(N)$[1].
> Therefore, for small $N$, MacKay's method can already be computed quite quickly, so the approximated method may not be necessary.
> Conversly, when $N$ is large, the computation of MacKay's method becomes slower, and we can use the approximated method to reduce the computational burden.
> Therefore, this approach can accommodate scenarios with both large and small values of $N$.
>
> Moreover, a decrease in accuracy on the MNIST dataset is also mentioned, which is mainly because the accuracies of the various pre-trained models on MNIST are actually quite similar.
> In this case, the estimation error of transferability introduces a significant decrease in correlation.
>
> [1] You, Kaichao, et al. "Ranking and tuning pre-trained models: A new paradigm for exploiting model hubs." Journal of Machine Learning Research 23.209 (2022): 1-47.

---

> > ### Comment · Reviewer_HHBr · 2025-01-24
> >
> > Thank you for your comments and revisions. My concerns are resolved.

---

> > > ### Author Response · Authors · 2025-01-25
> > >
> > > Thank you very much for your meticulous review and the valuable critiques on our manuscript. We greatly appreciate your help in improving our work!

---

### Review · Reviewer_6Yoo · 2025-01-01

**Summary Of Contributions:**

Spiking neural networks (SNNs) are in general hard for practitioners to train, so it would be desirable to have a selection of pre-trained SNN models from which to choose in order to perform the relatively easier procedure of transfer learning. When offered a selection of pre-trained models (PTMs), practitioners need a cheap way to assess which PTM is best for them, and while there are cheap and powerful ways to perform this assessment for standard PTMs, many of the developed methods are not applicable to SNN PTMs. The authors provide a new way to conduct such assessment of SNN PTMs for transfer learning tasks. The method builds upon previous work that approximates the (maximum log) marginal likelihood of a linear model mapping from the PTM-obtained features to their respective targets/labels. The main technical challenge that the authors overcome is the difficulty in assuming a linear model from SNN outputs (which, for a given input, is a 2D matrix corresponding to a sequence of representations) to targets---i.e. "how should we perform aggregation in the time-dimension such that a linear model can be assumed?". The authors propose averaging the 2D feature matrices in their time-dimension, and present an efficient way to approximate the maximum log marginal likelihood.

**Audience:**

Yes

**Broader Impact Concerns:**

No concerns as I do not believe a Broader Impact Statement is required.

**Claims And Evidence:**

Yes

**Requested Changes:**

1. In the paragraph between equations 1 and 2, the authors refer to the linear model's likelihood as the "prior distribution of the observed data"---I find this very peculiar. I have not heard the term "prior distribution of the observed data" before, but logically (to me anyways) the only distribution that makes any sense to correspond to this term would be $p(\mathcal{D})$---the marginal likelihood.
2. Equation 5 needs clarification regarding the use of $n$ to denote the network layer index. It took me a few reads to figure this out.
3. In the paragraph below equation 10, $\mathbf{f}$ seems to be missing an appropriate superscript (i.e. is it $\bar{\mathbf{f}}$ or $\mathbf{f}^{(t)}$?). Also the converse case of $t_i=0$ is not obvious to me; it would be nice to have more detail on this.
4. Between equations 3 and 4 the authors use the word “donate” when I believe it should be “denote”. The same goes for the paragraph before equation 6.
5. “Properties 1” should be “Property 1”.
6. Section 3 refers to Appendix B for proof of property 3---there is no appendix B, only subappendices of appendix A.
7. In appendix A.2.1 equation 19, I believe the equality between lines 2 and 3 might not be correct. Using equations 16, 17, and 18, I find the second line of equation 19 to be equal to $\mathcal{L}_1^* + \frac{\beta^*}{2}m^Tm\Bigl(1-\frac{1}{q^2}\Bigr)$, i.e. I cannot see how the $\frac{\beta^*}{2}\tilde{m}^T\tilde{m}$ term disappears. I could of course be wrong. Some more details on how this equality holds would be appreciated.
8. In appendix A.2.2 in the paragraph before equation 21, the authors refer to stacking $\mathbf{F}$ (the feature matrix) $q$ times in a row to reach [F, 0]. I imagine this is a copy-paste typo from the paragraph above.
9. In appendix A.2.2, the proof of theorem 1 looks right so long as the proof in A.2.1 is assumed to be correct. However, it requires heavy referral to [You et al., 2022], which makes it harder to parse on its own and harder to assess the correctness of. Some more details would be appreciated.

**Strengths And Weaknesses:**

**Strengths**:
- well-motivated problem,
- intuitive and pragmatic solution,
- good experimental results, including strong evidence that the proposed approximation method is sufficiently accurate.


**Weaknesses**:
- a number of typos and mistakes,
- no errorbars on any experimental results (I imagine this is due to limited computational budget rendering multiple SNN PTM training runs impractical),
- limited details in theoretical proofs, potentially a mistake in one of them too.

---

> ### Author Response · Authors · 2025-01-20
> **Response to the reviewer (1-4)**
>
> Dear reviewer,
>
> Thank you for your careful and kind review. Your comments are meaningful and helpful for our work. Here are our responses to your concerns.
>
>
> 1. no errorbars on any experimental results
>
> ### Response
> Thank you for pointing out this issue. We indeed faced limitations in computational resources during the training process, which made it impractical to run multiple SNN PTM training sessions. However, to provide a clearer indication of the error range in the results presented, we conducted several repeated experiments on the typical neuromorphic dataset DVS128 Gesture. For each PTM, we obtained eight sets of results by conducting the fine-tuning procedure with eight different random seeds, and we derived the error range of test set accuracy for this dataset based on these results. The results are shown in the table below. The results show that the error range is relatively small and have limited impact on the result of transferability assessment on this dataset. The corresponding results and figures have also been added to the revised manuscript in Appendix A.4 as Fig. 7 and Tab. 5. We hope these results can address your concerns to some extent.
>
> | MODEL             | ACC   | STD   |
> |-------------------|-------|-------|
> | Spiking MLP-12    | 0.306 | 0.009 |
> | Attention SNN     | 0.615 | 0.022 |
> | SEW ResNet-18     | 0.656 | 0.012 |
> | SEW ResNet-34     | 0.635 | 0.008 |
> | SEW ResNet-50     | 0.622 | 0.014 |
> | SEW ResNet-101    | 0.639 | 0.017 |
> | SEW ResNet-152    | 0.565 | 0.013 |
> | Spiking ResNet-18 | 0.523 | 0.012 |
> | Spiking ResNet-34 | 0.48  | 0.015 |
> | Spiking ResNet-50 | 0.497 | 0.012 |
>
>
> 2. In the paragraph between equations 1 and 2, the authors refer to the linear model's likelihood as the "prior distribution of the observed data"---I find this very peculiar. I have not heard the term "prior distribution of the observed data" before, but logically (to me anyways) the only distribution that makes any sense to correspond to this term would be $\mathcal{p}(\mathcal{D})$the marginal likelihood.
>
>
> ### Response
> Thank you for pointing out this issue. The expression here was indeed problematic. What we intended to convey is that the observed data $t_i$ is generated by transforming the input feature vector $f_i$ and adding noise with variance $ 1/\beta $. We have made the necessary revision in the manuscript, changing the wording to:\textit{ Furthermore, it assumes that the targets $ t_i $ are conditionally distributed according to a Gaussian likelihood$ t_i \sim \mathcal{N}(\theta^{\mathrm{T}} f_i, \beta^{-1}) $, where $ \beta^{-1}$ represents the noise variance.}
>
>
> 3. Equation 5 needs clarification regarding the use of to denote the network layer index. It took me a few reads to figure this out.
>
>
> ### Response
> Thank you for your advice. We have clarified the meaning of $n$ it in Equation 5.
>
>
> 4. In the paragraph below equation 10, seems to be missing an appropriate superscript (i.e. is it
>  $\mathbf{\bar{f}}$ or $\mathbf{f}^{(t)}$). Also the converse case of is not obvious to me; it would be nice to have more detail on this.
>
> ### Response
> (1) Thank you for your careful check. It should be $\bar{\mathbf{f}}$ in Equation 10.
>
> (2) The discussion for the case of $t_i = 0$ is indeed not obvious here. We have added a discussion on this situation in the revised manuscript on page 5.

---

> ### Author Response · Authors · 2025-01-20
> **Response to the reviewer（5-10）**
>
> 5. Between equations 3 and 4 the authors use the word “donate” when I believe it should be “denote”. The same goes for the paragraph before equation 6.
>
>
> ### Response
> Thank you for pointing this out. We have corrected them as "denote".
>
>
> 6. “Properties 1” should be “Property 1”.
>
>
> ### Response
> Thank you for pointing this out. We have changed “Properties” to “Property” in the revised manuscript.
>
>
> 7.Section 3 refers to Appendix B for proof of property 3---there is no appendix B, only subappendices of appendix A.
>
> ### Response
> Thank you for pointing out this issue, and we apologize for the mistake. We have corrected all the references to the appendix in the main text. We appreciate your careful review.
>
> 8. Problem with proof of property 1.
>
> ### Response
> Thank you for checking the proof in our manuscript. The problem with the previous proof in Appendix A.2.1 is that we made a mistake in Equation (16).
>
> The logarithm of maximum evidence should be
> \begin{equation}
>     \mathcal{L}_1^* = \frac{N}{2}\ln \beta^* + \frac{D}{2}\ln \alpha^* - \frac{N}{2}\ln {2\pi} -\frac{\beta^*}{2}\left\lVert \mathbf{F}m-\mathbf{t}\right\rVert ^2 -\frac{\alpha^*}{2}m^{\mathrm{T}}m-\frac{1}{2}\ln \det \mathbf{A}
> \end{equation}
> according to Equation (3).
> In fact, we make a mistake in writing the $\frac{\alpha^*}{2}m^{\mathrm{T}}m$ term as $\frac{\beta^*}{2}m^{\mathrm{T}}m$.
> So after correcting Equation (16), the corresponding item in Equation (20) becomes
> $ \frac{q^2\alpha^*}{2}\left(\frac{1}{q}m\right)^{\mathrm{T}}\left(\frac{1}{q}m\right)$ which exactly equals to $\frac{\alpha^*}{2}m^{\mathrm{T}}m$. Then the proof will be correct.
> We also add more details in the proof in the revised manuscript. Thank you again for your careful check.
>
>
> 9. In appendix A.2.2 in the paragraph before equation 21, the authors refer to stacking $\mathbf{F}$ (the feature matrix) $q$ times in a row to reach [F, 0]. I imagine this is a copy-paste typo from the paragraph above.
>
>
> Thank you for your check. We have changed the mistake in the revised manuscript.
>
>
> 10. In appendix A.2.2, the proof of theorem 1 looks right so long as the proof in A.2.1 is assumed to be correct. However, it requires heavy referral to [You et al., 2022], which makes it harder to parse on its own and harder to assess the correctness of. Some more details would be appreciate.
>
> Thank you for your advice. We have provided more detailed proof steps in the revised version.

---

> > ### Comment · Reviewer_6Yoo · 2025-01-24
> > **Sufficient Rebuttal**
> >
> > Many thanks for your detailed rebuttal and for implementing changes to reflect all of my concerns.
> >
> > My only remaining qualm is that the discussion of the case for $t_i=0$ in equation (10) (which I am very grateful that you have added) might be better placed in an appendix due to the density of equations---however this is just a suggestion.
> >
> > Regardless, I am very satisfied with each and every one of the changes and would be happy to recommend acceptance to TMLR.

---

> > > ### Author Response · Authors · 2025-01-25
> > >
> > > Dear reviewer,
> > >
> > > Thank you for your suggestion! We have moved the discussion regarding this scenario to Appendix A.2.3 in the latest version of the manuscript.
> > >
> > > Thank you very much for your recognition and meticulous review of our manuscript. We greatly appreciate your help in improving our work!

---

### Review · Reviewer_6bD1 · 2025-01-14

**Summary Of Contributions:**

This paper aims to assess the transferability of pre-trained spiking neural networks (SNNs), which many existing methods for deep learning models cannot be applied due to the difference in rate-based and spike-based architectures. The authors adapted an existing maximum evidence assessment method to SNNs by averaging the temporal dimension of the feature maps of the spike-based model. The authors evaluated their proposed method against two baseline methods on a range of static and spiking vision datasets. Moreover, the authors proposed an approximation approach to the iterative process used in the maximum evidence method. They showed that their method can achieve similar performance while reducing the overall computation time.

**Audience:**

Yes

**Broader Impact Concerns:**

I do not believe this work requires a broader impact statement.

**Claims And Evidence:**

Yes

**Requested Changes:**

Very minor comments and suggestions
- Can the authors clarify the x-axis of Figures 1, 2, 4 and 5 is the test accuracy? I don’t think they are currently labelled.
- I suggest adding a table in the main text that summarizes all the Kendall correlation coefficient values from Figures 1, 2, 4 and 5 so that the reader can easily compare all the methods in different dataset configurations.
- I believe Table 3 in Appendix A.3 is an important contribution of this work, which shows the time saved with the approximation method against MacKay’s method. I suggest adding the table, or at least referencing the table, in the main text.
- Some typos here and there, e.g. page 5: “equation equation 7”; page 6: “equation equation 3”; page 7: I believe it should be “we use PTMs from Fang et al., 2021.” (\citep vs \citet).

**Strengths And Weaknesses:**

Strengths
- Presents a method to estimate the transferability of SNNs, which is a relatively unexplored area.
- Comprehensive datasets for evaluation, including both static and spike-based vision datasets; as well as a toy dataset to evaluate the approximation and the original iterative methods of maximum evidence.
- Overall, the paper is well-written and easy to follow.

Weaknesses
- Only ResNet-based models are used in the analysis, which is noted in the limitation paragraph of the Discussion. However, I encourage the authors to add other SNN architectures, even simple models like SNN-based MLP, to the mix to show the generalizability of the method.
- Please see the Requested Changes section for minor suggestions.

---

> ### Author Response · Authors · 2025-01-20
>
> Dear reviewer,
>
> Thank you for your careful and kind review. Your comments are meaningful and helpful for our work. Here are our responses to your concerns.
>
> 1. Only ResNet-based models are used in the analysis, which is noted in the limitation paragraph of the Discussion. However, I encourage the authors to add other SNN architectures, even simple models like SNN-based MLP, to the mix to show the generalizability of the method.
>
> ### Response
> Thank you for your kind advice. In order to show the generalizability of our method, we add SNN-based MLP and attention-based SNN [1] in the experiments on the neuromorphic datasets. The newly added results are shown in Fig. 1, Fig. 4 and Tab. 1. The changes of Kendall coefficient after adding new SNN models are CIFAR10-DVS (0.93 $\xrightarrow{}$ 0.96), DVS128 Gesture (0.84 $\xrightarrow{}$ 0.78), N-Caltech101 (0.93 $\xrightarrow{}$ 0.87) and N-MNIST (0.86 $\xrightarrow{}$ 0.87) which are small. These results might be able to enhance the generalizability of the method.
>
> [1] Yao, Man, et al. "Attention spiking neural networks." IEEE transactions on pattern analysis and machine intelligence 45.8 (2023): 9393-9410.
>
>
> 2. Can the authors clarify the x-axis of Figures 1, 2, 4 and 5 is the test accuracy? I don’t think they are currently labelled.
> ### Response
> Thank you for your advice. The meaning of x-axis in Fig. 1, 2, 4 and 5 is indeed not clear. We add \textbf{assessed transferability scores (y-axis) vs. test set accuracy (x-axis)} in the caption of these figures to clarify the meaning of the x-axis. You can check it in the revised manuscript.
>
> 3. I suggest adding a table in the main text that summarizes all the Kendall correlation coefficient values from Figures 1, 2, 4 and 5 so that the reader can easily compare all the methods in different dataset configurations.
>
> ### Response
> Thank you for your advice. We have added the table in the main context as shown below. You can also check it in the last subsection of section 4 (section Results) as Table 2 in the revised manuscript page 12.
>
> |DATASET  | CIFAR10-DVS | DVS128 Gesture | N-MNIST | N-Caltech101 | CIFAR10 | CIFAR100 | Caltech101 | MNIST |
> |------|------|------|------|------|------|------|------|------|
> | NCE | 0.60 | 0.16 | -0.02 | 0.29 | 0.27 | 0.57 | -0.11 | 0.57 |
> |LEEP | 0.56 | 0.20 | 0.20 | 0.51 | 0.00  | -0.07 | -0.11 | -0.29 |
> |MEAF(+MacKay) | 0.96 | 0.78 | 0.87 | 0.87 | 0.64 | 0.64 | 0.76 | 0.71 |
> |MEAF(+Approximated) | 0.96 | 0.56 | 0.87 | 0.87 | 0.64 | 0.64 | 0.76 | 0.57 |
>
>
> 4. I believe Table 3 in Appendix A.3 is an important contribution of this work, which shows the time saved with the approximation method against MacKay’s method. I suggest adding the table, or at least referencing the table, in the main text.
>
> ### Response
> Thank you for your advice. We have moved this table as Table 1 in the last subsection of section 4 (section Results). You can see it in the revised manuscript on page 11.
>
>
> 5. Some typos here and there, e.g. page 5: “equation equation 7”; page 6: “equation equation 3”; page 7: I believe it should be “we use PTMs from Fang et al., 2021.” (\textbackslash citep vs \textbackslash citet).
>
> ### Response
> Thank you for your careful check! We have corrected the above errors in the revised manuscript.

---

> > ### Comment · Reviewer_6bD1 · 2025-01-24
> >
> > I thank the authors for their detailed responses which have addressed my main questions and concerns.

---

> > > ### Author Response · Authors · 2025-01-25
> > >
> > > Thank you very much for your meticulous review of our manuscript. We greatly appreciate your help in improving our work!

---

### Decision · Action_Editor_sBDG · 2025-02-23

**Recommendation:** Accept with minor revision

**Comment:**

The paper is generally well written but I still have a few minor comments to improve the quality of the paper. I encourage the authors to improve this, although the decision of the paper does not depend on these changes.

1. The matrix vector notation is mixed in Eq. 2-4, e.g., t is a bold-faced vector while $f_i$, $\theta$, $m$ etc. are not bold-faced despite being vectors.

2. I felt that it is better to write “invariance of Mackay’s method” rather than “invariant” (same thing for all the properties).

3. Font size in figures is too small. In general, the font size in the figures should be only slightly smaller than the main text. Please increase marker size. Use rid lines, so that it is easier to see the differences. Font in Table 1 and 2 can also be fixed easily (eg, use abbreviation for the headings).

4. Reference formatting can be improved, for example, "r-stpd" is "R-Stpd" in the original paper, "ssn" should be "SSN", and in Kunster et al. 2019, Fisher does not have a capitalized F. Also, "bayesian" should be "Bayesian". I also find it amusing that in one of the reference my name is written wrong (it is "Mohammad Emtiyaz Khan", not "Khan Mohammad Emtiyaz") but perhaps this is propagated through Google Scholar, so I don't blame the authors :)

5. I think that the citation for Kunstner et al. is not appropriate. This paper is for Fisher matrix, not on model evidence, from what I remember. So it might be better to remove it from this particular location.

These are only minor changes and will improve the formatting of the paper.

**Audience:**

The paper will of interest to the researchers working on spiking neural networks. I am not sure how large this community is within machine learning but I am sure folks interested in neuroscience will benefit from such works. In this sense, I believe the paper is suitable to be published in TMLR.

**Claims And Evidence:**

The paper proposes a new method to approximate model evidence for spiking neural networks. The methodologies used for Pre-Trained Models is adopted and a simple approximation method is proposed which does not require taking multiple iterations. The usefulness of the new approximation is shown on multiple models. The reviewers are satisfied with the correctness of the approach and the quality of experiments. Even though the methods builds on existing approaches, the reviewer recognize the usefulness of the approach and they all unanimously agree to accept the paper.

---

> ### Author Response · Authors · 2025-03-10
> **Changes in the revised manuscript**
>
> Dear Action Editor,
>
> Thank you for your recommendation and comments on some minor revisions. We have made the modifications and submitted a revised version of the manuscript.
>
> However, we would like to bring to your attention an issue. In preparation of the camera-ready version of our manuscript, we discovered an error in loading incorrect weights and mishandling the experimental data of the static datasets. We sincerely apologize for this error.
>
> In the revision, we have corrected the relevant experimental results. These corrections affect only Figure 2, Figure 5, Table 2, and Appendix Table 6. We checked that the corrected experimental results do not impact the corresponding conclusions, namely that MEAF demonstrates better performance in SNN transferability assessment compared to LEEP and NCE. This can be observed in the revised Table 2.
>
> | **DATASET**   | **C10-DVS** | **DVS128** | **N-MNIST** | **N-C101** | **C10** | **C100** | **C101** | **MNIST** | **AVERAGE** |
> |---------------|-------------|------------|-------------|------------|---------|----------|----------|-----------|-------------|
> | NCE           | 0.60        | 0.16       | -0.02       | 0.29       | **0.86**| **0.71** | 0.50     | 0.29      | 0.42        |
> | LEEP          | 0.56        | 0.20       | 0.20        | 0.51       | **0.86**| **0.71** | **0.64** | 0.29      | 0.50        |
> | MEAF (MacKay)  | **0.96**    | **0.78**   | **0.87**    | **0.87**   | **0.86**| **0.71** | 0.50     | **0.79**  | **0.79**    |
> | MEAF (Approx)  | **0.96**    | 0.56       | **0.87**    | **0.87**   | **0.86**| **0.71** | 0.43     | 0.64      | 0.74        |
>
> Nevertheless, we are sorry for the error and are willing to undergo re-evaluation or engage in further discussions with the reviewers if you deem it necessary.

---

> > ### Comment · Action_Editor_sBDG · 2025-03-18
> > **The claim needs to be revised.**
> >
> > Dear authors,
> >
> > Thanks for letting us know about the error. We all appreciate the honesty.
> >
> > I discussed these with the reviewers. They are still in favor of acceptance but would like to change your claim to clearly state the method outperforms NCE and LEEP only for neuromorphic datasets, but is comparable on static datasets. The only static dataset where you do well is the MNIST which is not conclusive. With the relaxed claim, you should also discuss why the method shows improvement in neuromorphic datasets but not static datasets, at least qualitatively, but ideally through some hypothesis analysis experiments. Without these changes, the reviewers do not want to accept because it will be against TMLR's criteria of claims being supported by results.
> >
> > Please do understand that if these changes are not made, the paper might be rejected. We look forward to the revision. Please do color code the new parts so that the reviewers can check these changes.

---

> > > ### Author Response · Authors · 2025-03-23
> > > **Changes in the revision**
> > >
> > > Dear Action Editor,
> > >
> > > Thanks for your comments and the opportunity to revise our manuscript.
> > > We have made the required changes in the revised manuscript, with all modifications marked in blue in the newly submitted revision.
> > > Here we summarize our changes in two aspects.
> > >
> > >
> > > ###  Modification of the claim and addition of qualitative explanation
> > > We have modified the claim to "MEAF performs better on neurmorphic datasets while achieving comparable results on static datasets" in the revised manuscript.
> > > A discussion was added after the experimental results in section 4.2 and 4.3, to explain our method (MEAF) outperforms LEEP and NCE on the neuromorphic datasets: Because neuromorphic datasets have an additional temporal dimension, SNNs would extract a more complicated feature map. Therefore, assessing transferability based on the feature map (e.g., MEAF) could leverage more information than on the outputs, which indicates our method outperforms output-based LEEP and NCE.  On the other hand, static datasets have no temporal information, thus SNNs act like ANNs, making LEEP and NCE perform as if they wrere on ANNs.
> > >
> > >
> > > ###  The results of a new metric for compare MEAF, LEEP and NCE
> > > For a more comprehensive comparison among MEAF, LEEP and NCE, we also added the results of Pearson correlation coefficient in the revised manuscript, which is used in the LEEP and NCE paper as the main metric.
> > > The added results also validate our claims that MEAF outperforms LEEP and NCE on neuromorphic datasets, and achieves comparable performance with LEEP and NCE on static datasets.
> > > The results are given in Table 1, 2, and 3 in the revised manuscript, and are also summerized and shown as below:
> > >
> > > ### Pearson Correlation Coefficients of NCE, LEEP, and MEAF on Neuromorphic and Static Datasets
> > >
> > > | Dataset | C10-DVS (Neuro) | DVS128 (Neuro) | N-MNIST (Neuro) | N-C101 (Neuro) | C10 (Static) | C100 (Static) | C101 (Static) | MNIST (Static) |
> > > |--------|------------------------|-----------------------|------------------------|-----------------------|------------------|-----------------|------------------|-----------------|
> > > | NCE    | 0.67                   | 0.08                  | 0.11                   | 0.63                  | 0.89             | 0.87            | 0.62             | 0.37            |
> > > | LEEP   | 0.76                   | 0.31                  | 0.24                   | 0.57                  | 0.90             | 0.90            | **0.80**         | 0.38            |
> > > | MEAF (+MacKay)   | **0.99**               | **0.91**              | **0.97**               | 0.96              | 0.95         | **0.91**        | 0.64             | 0.83        |
> > > | MEAF (+Appox)  | **0.99** | 0.87 | **0.97** |  **0.98** |  **0.96**  |  **0.91**  | 0.63 | **0.86** |
> > >
> > > Under the metric of Pearson correlation coefficient, MEAF outperforms LEEP and NCE on all the neuromorphic datasets, and have better performance on 3 of the 4 static datasets.
> > > We hope this could give a more comprehensive comparison of the 3 methods.

---

> > > > ### Comment · Action_Editor_sBDG · 2025-04-15
> > > > **Thanks for fixing the error**
> > > >
> > > > Dear authors,
> > > >
> > > > Thanks for fixing the error. I believe it is okay to accept the paper in this new form. Feel free to add a new version where blue parts are changed to the normal font colors.